# Economies of Open Intelligence:
# Tracing Power & Participation in the Model Ecosystem

## Abstract

Since 2019, the Hugging Face Model Hub has been the primary global platform for sharing open weight AI models. By releasing a dataset of the complete history of weekly model downloads (June 2020–August 2025) alongside model metadata, we provide the most rigorous examination to-date of download-based concentration dynamics and evolving characteristics in the open model ecosystem. Our analysis spans 851,000 models, over 200 aggregated attributes per model, and 2.2B downloads, establishing persistent scientific infrastructure for measuring how AI capability, influence, and participation diffuse across the global research and deployment landscape. We document a fundamental rebalancing of usage share: US open-weight industry dominance by Google, Meta, and OpenAI has declined sharply in favor of unaffiliated developers, community organizations, and, as of 2025, Chinese industry, with DeepSeek and Qwen models potentially heralding a new consolidation in download-based usage. We identify statistically significant shifts in model properties, including a $17\times$ increase in average model size, rapid growth in multimodal generation ($3.4\times$), quantization ($5\times$), and mixture-of-experts architectures ($7\times$), alongside concerning declines in training-data transparency, with open-weight models lacking sufficient training-data disclosure surpassing models that satisfy core open-source AI documentation criteria for the first time in 2025. We expose a new layer of developer intermediaries that has emerged, focused on quantizing and adapting base models for both efficiency and artistic expression. To enable continued research and oversight, we release the complete dataset with an interactive dashboard for real-time monitoring of usage concentration, innovation diffusion, and evolving properties in the open model ecosystem.

🏆 **Dashboard** *Link omitted for double-blind review*

## 1 Introduction

The concentration of power in artificial intelligence across computational resources (Lehdonvirta et al., 2024a), training data (Buolamwini & Gebru, 2018; Longpre et al., 2024; Longpre et al.), and model development, has emerged as a central focus for the fairness, safety, and control of the AI economy (Hopkins et al., 2025). Understanding where influence concentrates, how it shifts over time, and which actors control the development and distribution of AI models is essential for effective oversight and equitable access (Crawford, 2021a; Noble, 2018). While prior work has examined the broader AI supply chain (Hopkins et al., 2025), the topology of Hugging Face model relationships (Laufer et al., 2025a; Horwitz et al., 2025b), or the implications of economic consolidation in the *closed* model AI market (Korinek & Vipra, 2025; Vipra & Korinek, 2023; Vipra & West, 2023; Nagle & Yue, 2025), no study has systematically traced how download-based usage concentrates and diffuses in the *open* model ecosystem over time, nor rigorously examined which model characteristics are waxing or waning in adoption.

The Hugging Face Model Hub provides a unique window into these dynamics. What began as a mode of distributing the open PyTorch formats of BERT and GPT-2 models in 2019 has evolved into the primary global platform for sharing open weight AI models, now hosting 2M+ models with 1.7B unique, cumulative

downloads.[1] This platform has become central to the international adoption and distribution of open weights models for both research and production use, spanning the full range of general-purpose AI tasks, modalities (text, speech, image, video, tabular), and languages. As such, it offers the most comprehensive view available into the evolution of the open AI model ecosystem, especially the dynamics of control, consolidation, and platform-mediated usage.

In this work, we aggregate and publicly release the complete historical download logs for the Hugging Face Model Hub spanning June 2020 to August 2025, merged with extensive metadata on model training methods, architectures, modalities, languages, developer country of origin, documentation quality, and access restrictions. This represents the largest and most rigorous study of open model usage to date, encompassing detailed analysis of 851,000 models (97.6% of all downloads) with additional collected metadata, often absent from automated records. Using temporal download patterns as the closest available proxy for model adoption, we trace long-term shifts and measure the concentration of download-based usage share across models, developers, and countries using established economic metrics including the Herfindahl-Hirschman Index (HHI) and Gini coefficient. We do not claim that downloads directly measure deployment, API invocations, revenue, or downstream task usage; instead, we interpret them as a large-scale, longitudinal proxy for relative attention and reuse on the dominant open-model distribution platform.

**Our central contribution is to the *science of science*: we introduce new instrumentation that transforms a fast-moving, previously unmeasurable domain, the open AI model ecosystem, into a tractable scientific object amenable to longitudinal analysis.** We build automated ecosystem monitoring infrastructure using AI-assisted annotation to continuously track model lineage, architectures, training methods, and provenance at scale, creating persistent scientific infrastructure analogous to telescope surveys or genomic databases, but for AI systems. We adapt established economic concentration measures, specialized variants of HHI and Gini coefficients, into ecosystem health metrics tailored to model-, developer-, and country-level usage dynamics, enabling rigorous, quantitative analysis of concentration, dependency fragility, and innovation diffusion patterns where previously only anecdotal claims were possible.

Our analysis reveals several critical findings for the open AI economy:

1. **Declining US industry usage share, rising influence of China and unaffiliated developers**: Concentration measures show a steep decline in US industry's usage share by Google, Meta, and OpenAI, starting in 2022. Download-based influence has diffused to unaffiliated users, community developers, and recently to Chinese industry, with DeepSeek and Qwen models potentially ushering in a new consolidation of usage share led by Chinese developers.

2. **Shift toward larger, multimodal, and computationally efficient architectures**: Models downloaded in 2025 grew to 17× the average 2020 parameter count, and exhibited more multimodal and video generation capabilities (3.4× increase). To accommodate these larger sizes, we see rising use of quantization techniques (5×), parameter-efficient finetuning (1.4×), and mixture-of-experts architectures (7×).

3. **Emergence of a developer intermediary layer**: Organizations that quantize, finetune, re-package, or build artistic adapters for major base models have surged in popularity, constructing a fundamental new layer of model intermediaries between base model creators and end users.

4. **Sharp decline in training-data transparency**: Data transparency has deteriorated significantly, with the proportion of downloads for models disclosing available training data falling from 79.3% (2022) to just 39% (2025). For the first time in 2025, downloads of open-weight models lacking sufficient training-data disclosure surpassed downloads of models meeting the Open Source Initiative's training-data documentation criterion for open-source AI[2].

5. **Public release of data and live monitoring dashboard:**[3] To enable continued research, transparency, and oversight, we release the complete dataset alongside an interactive dashboard for

---

[1] Note: we use a more precise measure of downloads than the raw counts reported on the Hub (see Section 2).

[2] https://opensource.org/ai/open-source-ai-definition

[3] Dashboard link omitted for double-blind review. The live public dashboard gives slightly different results as it uses publicly available downloads data, which is less precise than the internal, deduplicated data used in this paper.

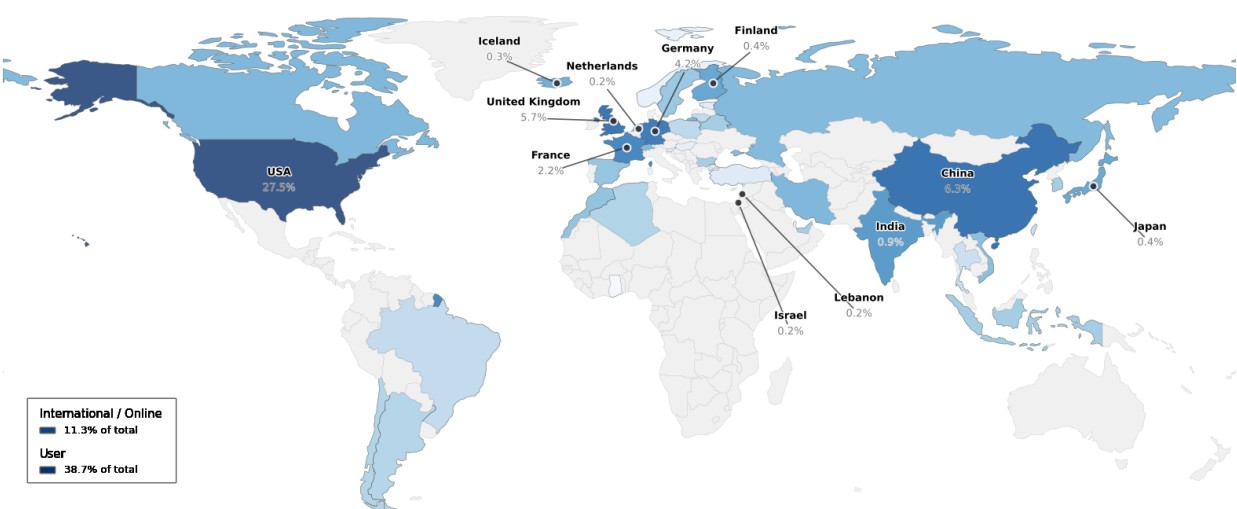

| | Top Countries | | | Top Developers | | | Top Models | | |
|---|---|---|---|---|---|---|---|---|---|
| **All-time** | Unaffil. User | 👤 | 38.7% | Google | 🇺🇸 G | 6.5% | adetailer | 👤 ✿ | 2.9% |
| | USA | 🇺🇸 | 27.5% | stable-diffusion | 🇬🇧 s. | 5.4% | bert-base-uncased | 🇺🇸 G ✿ | 2.9% |
| | Internat./Online | 🌐 | 11.3% | Bingsu | 👤 | 5.3% | yolo-world-mirror | 👤 ✿ | 2.4% |
| | China | 🇨🇳 | 6.3% | lllyasviel | 👤 | 4.7% | clip-vit-large-patch14 | 🇺🇸 🌀 ✿ | 2.1% |
| | UK | 🇬🇧 | 5.7% | Facebook | 🇺🇸 ∞ | 4.6% | stable-diffusion-xl-base-1.0 | 🇬🇧 s. 🖼 | 1.4% |
| | Germany | 🇩🇪 | 4.2% | lmstudio-community | 🌐 📦 | 4.0% | distilbert-base-uncased | 🇺🇸 🤗 ✿ | 1.1% |
| | France | 🇫🇷 | 2.2% | OpenAI | 🇺🇸 🌀 | 3.6% | fav_models | 👤 ◆ | 0.9% |
| | India | 🇮🇳 | 0.9% | sd-concepts-library | 🌐 🏛 | 2.8% | misc | 👤 ◆ | 0.9% |
| | Finland | ➕ | 0.4% | timm | 🇺🇸 timm | 2.3% | dependencies | 👤 ◆ | 0.8% |
| | Japan | 🔴 | 0.4% | deepseek-ai | 🇨🇳 🐋 | 2.3% | stable-diffusion | 🇩🇪 🏛 🖼 | 0.8% |
| **Aug. 2024 – Aug. 2025** | Unaffil. User | 👤 | 33.8% | lmstudio-community | 🌐 📦 | 16.4% | DeepSeek-R1 | 🇨🇳 🐋 📝 | 2.8% |
| | Internat./Online | 🌐 | 23.8% | deepseek-ai | 🇨🇳 🐋 | 9.6% | stable-diffusion-v1-5 | 🇬🇧 s. 🖼 | 2.4% |
| | China | 🇨🇳 | 17.1% | comfy | 🇺🇸 G | 5.4% | flux_text_encoders | 🇺🇸 G ✿ | 1.2% |
| | USA | 🇺🇸 | 15.8% | Qwen | 🇨🇳 🦅 | 4.6% | Wan_2.1_ComfyUI_repackaged | 🇺🇸 G 🖼 | 1.2% |
| | UK | 🇬🇧 | 3.6% | stable-diffusion | 🇬🇧 s. | 3.6% | stable-diffusion-v1-5-archive | 🇺🇸 G 🖼 | 1.1% |
| | India | 🇮🇳 | 3.4% | strangerzonehf | 🇮🇳 🖼 | 3.4% | Llama-3.2-1B-Instruct-q4f16_1-MLC | 🌐 🐙 📝 | 1.0% |
| | France | 🇫🇷 | 1.0% | alvdansen | 👤 | 3.1% | DeepSeek-V3 | 🇨🇳 🐋 📝 | 1.0% |
| | Germany | 🇩🇪 | 0.4% | mlx-community | 🌐 MLX | 2.6% | Qwen2.5-0.5B-Instruct-q4f16_1-MLC | 🌐 🐙 📝 | 0.9% |
| | Singapore | 🇸🇬 | 0.2% | Shakker-Labs | 🇺🇸 ● | 2.5% | clip-vit-base-patch32-ONNX | 🌐 ONNX ✿ | 0.8% |
| | Switzerland | 🇨🇭 | 0.2% | openfree | 👤 | 2.2% | Janus-Pro-7B | 🇨🇳 🐋 📝 | 0.8% |

Figure 1: **Top:** The Top 12 Nation Map ranked by the all-time 🤗 downloads of their models.
**Bottom:** The top 10 🤗 downloads Leaderboard for countries, developers, and models, with their download percentages. Both the map and the leaderboard use Rolling Window Filter to mitigate inauthentic downloads. The ***All-time*** section reflects all time downloads, whereas the ***Aug. 2024 - Aug. 2025*** reflects all downloads for models created within the last year (August 2024 to August 2025). Symbols indicate details about the model: ✿ = embedding and classification models; 📝 = text generation, 🖼 = image generation, 🎙 = speech generation, 🎞 = video generation, 🌐 = international/online organization, 👤 = unaffiliated user.

> real-time monitoring of usage concentration, participation shifts, and evolving model properties in the open AI ecosystem.

These findings provide critical empirical grounding for policy discussions around AI governance, usage concentration, and the preservation of open and equitable access to AI capabilities.

## 2 Experimental Methodology

Our temporal usage analysis of the open model ecosystem is made feasible by merging metadata from several sources: Hugging Face's entire history of model downloads, automatic crawls of the Model Hub directories, and manually collected annotations: the derivation tree from other models, architectures, input/output

modalities, training/inference methods, languages, transparency, documentation, and access restrictions. Full data collection details are provided in Appendix B.

**Usage Data.** For the usage data, we aggregate model downloads, at a weekly cadence, from the Hugging Face Model Hub, since they first started recording in June 2020. Hugging Face's application team reports download counts using a privacy-preserving de-duplication rule: at most one download is counted per user per model per day. This rule reduces repeated automated pulls while preserving repeated activity across days, which is often the strongest observable signal available for ongoing use. We further filter out models with $< 200$ total downloads, to not over-count repetitive automatic processes, or models that are not frequently used beyond the developer. These filters yield 851k models of all 1.88M available on the Hub, however, this sample accounts for 97.6% of all downloads. We denote this as the **Population Sample**. From this population, we conduct a stratified sample of the top 200 most downloaded models for each of the 265 weeks between June 2020 and August 2025. This yields 2875 models, comprising 49.6% of all downloads, which we call the **Head Sample**. We collect significantly more detailed metadata using trained human annotators about the model sizes, languages, derivations, architectures, training methods, and data sources, which Hugging Face metadata often misses.

Note that there are few sources for open model *usage*, and few are systematically reliable. Downloads do not directly measure deployments, API calls, model invocations, revenue, or downstream citations. For instance, we investigated the well-known OpenRouter, however its model selection and user-base is limited, and the metrics are skewed towards models that do not have APIs available elsewhere.[4] Other complementary adoption signals, such as scholarly mentions, benchmark submissions, or GitHub dependencies, capture important but narrower slices of the ecosystem. We therefore treat Hugging Face downloads as a platform-mediated proxy for relative attention and reuse, not as a direct measurement of economic value or total real-world deployment. After de-duplication and filtering, it is nevertheless the most comprehensive longitudinal signal currently available for open model adoption. Even so, raw counts show a disproportionate number of downloads for pre-2023 models. But many of these downloads are driven by outdated automatic software that continuously loads models without any meaningful usage.[5] This can lead to skewed estimates of usage and popularity. To account for this, our main analysis uses a Rolling Window Filter metric, where downloads are only counted if the model was created within one year of the download. We posit this filter isolates more authentic popularity and usage trends, before the download signal is overcome by automatic processes that do not actually use the models. Typically, more performant or efficient versions of models are available well within one year. In Appendix D, we add sensitivity analyses with no recency cap and a stricter 6-month cap. The qualitative pattern of declining early concentration followed by renewed concentration in 2025 remains, while the no-cap analysis visibly overweights legacy BERT-, CLIP-, and embedding-based models.

Lastly, there is a choice on how to attribute credit when a model A is downloaded, but it was adapted (finetuned, quantized, etc) from a base model B. Both choices are valid though they offer different assumptions. For some analyses we use the latter, which we refer to as Recursive Model Attribution, applying credit to the base models and their developers. Formally, we construct a directed derivation graph from model $A$ to its listed parent model(s). For a downloaded model, we recursively follow parent links until reaching a root model with no known parent and attribute the download to that root model's author and country. If multiple parents are listed, we use the first documented parent, matching the order in the model card or annotation record; if no parent is documented or the parent is unavailable in our public-model snapshot, we retain the downloaded model's own author and country.

**Model Characteristic Data.** To ascertain trends in model usage, we also collect significant metadata that is not already available for models on the Hugging Face Hub. We curated detailed instructions, and hired expert annotators, compensated at 25$ per hour. Before completing real annotations, annotators completed multiple rounds of training guided by expert annotators. After annotation, an expert NLP researcher reviewed a random sample of approximately 10% of all annotator labels; all fields were at least 90% accurate, with model architecture the lowest-scoring category. To ascertain model sizes, we use the Model Hub `safetensors` field where available, and otherwise estimate the parameter count from the models' file sizes. We employed

---

[4]`https://openrouter.ai/rankings`
[5]For instance vLLM automatically downloads OPT-125M as part of its CI/CD despite minimal actual usage (see `https://github.com/vllm-project/vllm/issues/7053`).

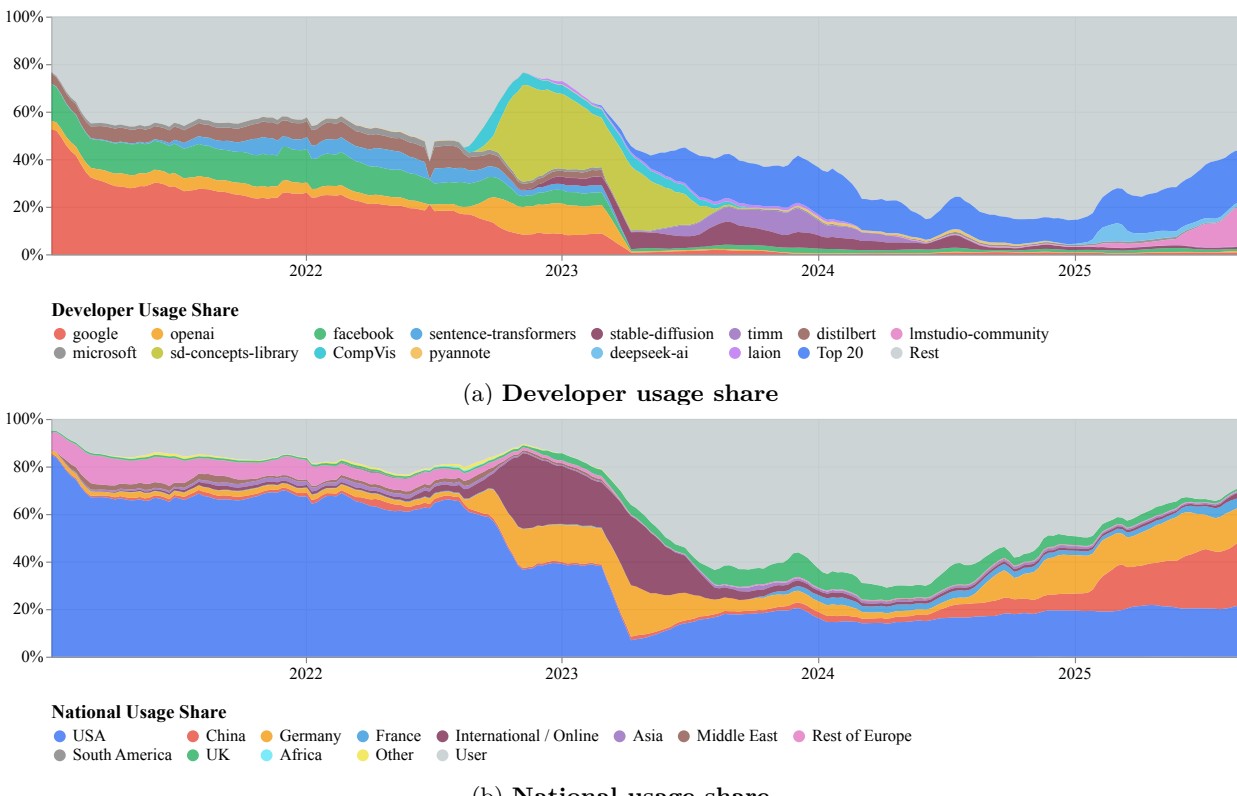

(a) **Developer usage share**

(b) **National usage share**

Figure 2: **Top:** Developer download-based usage share over time, using the Rolling Window Filter and applying Recursive Model Attribution. Where Google, Meta, and OpenAI once dominated usage share (2021–2024), their influence has subsided as developers beyond the Top 20 have gradually increased from 20% to >50% share. **Bottom:** National download-based usage share over time, using the Rolling Window Filter and applying Recursive Model Attribution. Where the US and Europe once dominated usage share (2021–2023), now Users, China, and Germany have become prominent contributors. Both plots use a 1-year Rolling Window Filter to better estimate authentic usage.

RANSAC (Random Sample Consensus) regression, a robust estimator that automatically identifies and excludes outliers arising from varying model architectures and compression ratios. This method obtained a validation $R^2 = 0.86$, and increased model size coverage from 23.6% to 98% of models. Full details of the annotation process are provided in Appendix E. Because the **Head Sample** is intentionally popularity-weighted, all annotated attribute trends should be read as download-weighted trends among high-usage models rather than unweighted prevalence across every model on the Hub.

**Usage Concentration Metrics.** The field of economics has developed concentration metrics to measure distributional inequality in economic systems. Most widely recognized is the Herfindahl-Hirschman Index (HHI). Calculated as the sum of squared shares within a distribution, it ranges from near zero (perfect competition) to 10,000 (monopoly); we rescale it from 0 to 1 Hirschman (1964); Rhoades (1993). For a complementary perspective, the Gini coefficient, ranging from 0 (perfect equality) to 1 (maximum inequality), measures the cumulative deviation from perfect equality across the entire distribution Gini (1912); Sen (1997). While both metrics increase with concentration, they capture distinct phenomena: HHI emphasizes the largest participants at the head of the distribution, whereas the Gini coefficient evaluates the equality across all participants equally Kwoka Jr (1985). In our setting, the "shares" are download-based usage shares, not validated revenue or deployment shares. We use the economic metrics because they compactly summarize whether usage is concentrated among a few models, developers, or countries, while avoiding claims that downloads are equivalent to economic power.

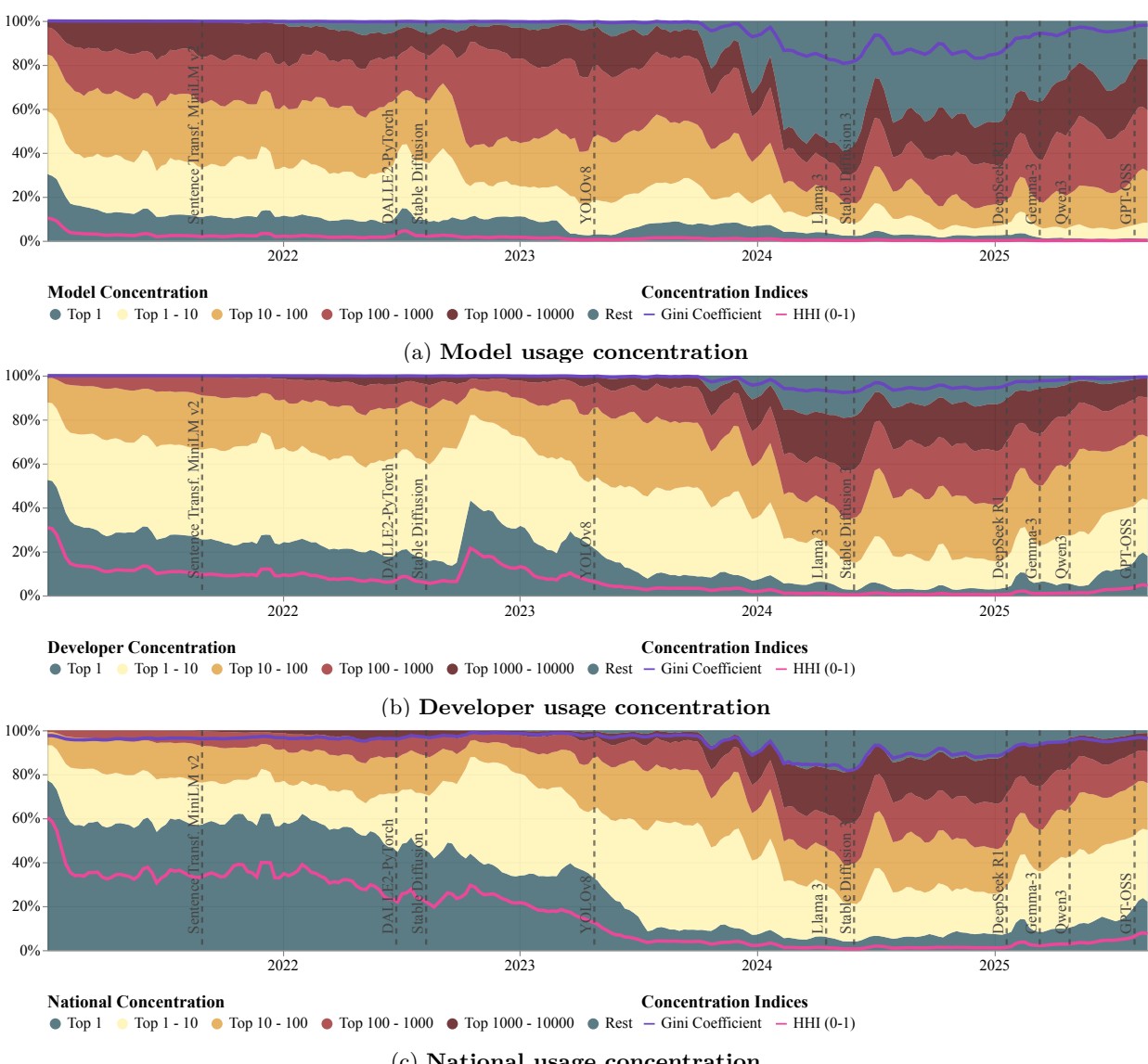

(a) **Model usage concentration**

(b) **Developer usage concentration**

(c) **National usage concentration**

Figure 3: In each plot we measure the share of downloads allocated to ranked segments of the open model ecosystem. Concentration metrics are also displayed in purple (the Gini coefficient) and pink (the Herfindahl-Hirschman Index from 0–1). **Across levels of abstraction (model, developer, nation) usage concentration first declined significantly, but has started to rise again in 2025.**

# 3   Usage Concentration: Open Models, Developers, & Nations

To gain an insight into concentration in the open model ecosystem, we trace download-based usage share at the model, developer, and national levels. Where prior work has mapped the concentration of computational resources (Lehdonvirta et al., 2024a), the concentration and geographical representation of training data (Longpre et al., 2024; Longpre et al.), and assessed static views of the Hugging Face model graph (Laufer et al., 2025a; Horwitz et al., 2025b), this is the first work to comprehensively map open model *usage* over the full historical time range that Hugging Face has recorded. This enables us to trace patterns of usage concentration, and their evolution. We first examine the all-time snapshot of the ecosystem (upper Figure 1), then explore three *periods* of model development, traced in the temporal breakdowns of usage share (Figure 2) and usage concentration (Figure 3).

**An All-Time Snapshot of the Ecosystem.** First, in Figure 1 we take a snapshot of the current open model ecosystem, from two perspectives: ***All-time*** and authentic usage from the ***Aug 2024 - Aug 2025***. The global map and ***All-time*** segment of the Leaderboard show download shares from June 2020 - Sept

2025, filtered by the Rolling Window Filter to mitigate inauthentic usage. The interactive dashboard allows readers to see leaderboard statistics both with and without these filters.

Across all-time we see the USA, Western Europe (UK, Germany, France), have dominated development, through mostly industry organizations: Google, OpenAI, Meta, HuggingFace's Sentence-Transformers, and Stable-Diffusion. American text and image embedding/classification models, based on CLIP (Radford et al., 2021), BERT (Devlin et al., 2019), YOLO (Redmon et al., 2016), and their variants account for many of the all-time downloads.

**The Foundation Embeddings Era (before late 2022)** Prior to late 2022, the open model ecosystem was characterized by high usage concentration and geographic consolidation. The top three US-based organizations, Google, Meta, and OpenAI, commanded between 40–60% of cumulative downloads, while the USA alone represented over 60% of national usage share. Figure 3 shows that in this period, the top 100 developers comprised over 90% of all downloads. HHI measures at the national and developer level were highest during this period, beginning around 0.3 and 0.6 in 2021. This era was dominated by a homogeneous consolidation around certain types of models: encoder-based architectures, with text encoder-only transformers, and embedding/classification heads comprising 76.8% and 75.2% of all downloads, respectively (see Table 1). Models were relatively small (average 217M parameters) with high transparency standards: 79.3% disclosed their training data availability. The technological focus centered on foundational capabilities like BERT (Devlin et al., 2019), CLIP (Radford et al., 2021), DistilBERT (Sanh et al., 2020), and sentence transformers (Reimers & Gurevych, 2019) that would later enable more sophisticated applications.

**The Generative Diffusion Period (late 2022 to early 2024)** This period witnessed a dramatic democratization of AI development following Stable Diffusion's release (Rombach et al., 2022), a *diffusion* both of generative architectures and of developer participation. Diffusion-based networks spike to 20% of all downloads, compared to $< 1\%$ a year prior. This sparked the other defining feature of this period: usage concentration plummeted as international and online organizations, followed by unaffiliated users, began developing on these models. Entities like CompVis, sd-concepts-library, and communities that develop text and image adapters (Houlsby et al., 2019) surged to prominence, collectively capturing notable portions of the download share (Figure 2). Many of these user communities began to form around libraries of artistic styles and renditions that could be applied to Stable Diffusion models.

Figure 3 shows this period is marked predominantly by a diffusion of usage share. Measures of concentration, especially HHI and the Top-10 share, begin declining precipitously, especially in early 2024. Notably, the "International/Online" category in national share experienced its largest spike, and the "Rest" category of developers grew from ∼20% to over 50% of usage share. Simultaneously, the popularity of OpenAI models rose significantly with wide interest in their image embedding and speech generation models, CLIP-VIT-Large (Radford et al., 2021), and Whisper (Radford et al., 2023). This era marked a fundamental shift from industry-led development to grassroots innovation, as thousands of community members created LoRA adapters, textual inversions, and fine-tuned variants. The barriers to AI development lowered substantially, enabling individual users and small collectives to meaningfully participate in the ecosystem.

**The Sino-Multimodal Period (late 2024 to the present)** The most recent period represents a plurality of changes: a fundamental re-balancing of download-based influence in the open ecosystem, a renewed rise in greater usage concentration, and a shift to larger but more quantized, multimodal reasoning models. China's share of downloads surged to 17.1% in the recent year, surpassing the collective of American model developers for the first time in our download-based attribution. These advances are driven primarily by DeepSeek and Qwen's reasoning models, with their developers capturing 14% of recent downloads alone. At the same time, Figure 3 shows that usage concentration metrics have suddenly started to rise in 2025, as individual Chinese models and developers consolidate greater download numbers.

Model complexity has also increased dramatically. Table 1 shows mean downloaded model size grew to 20.8B parameters (17× increase), mainly due to advances in quantization and wider adoption of mixture-of-expert architectures (where a subset of parameters are active at a time). Similarly, adoption of multimodal generation and video generation models, such as WanAI's Wan2.1 series, each rapidly expanded by 3.4×. However, this sophistication came with reduced transparency: only 39% of model downloads are for systems that disclose available training data (down from 79.3%). A new infrastructure layer has emerged, with organizations such

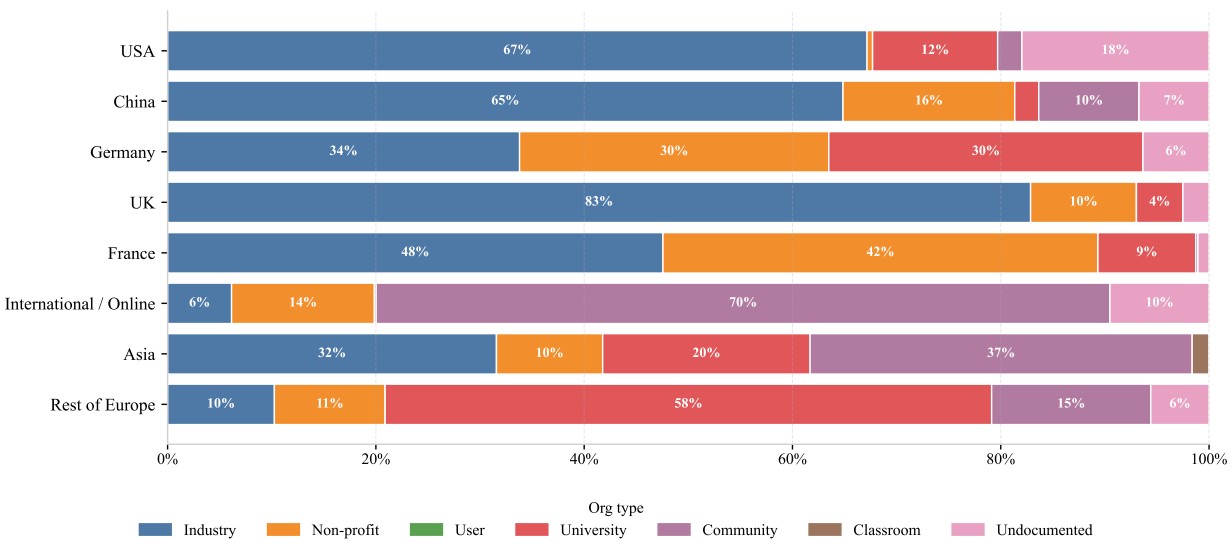

Figure 4: The proportion of downloads allocated by developer organization type in each country. **We find US, China, and UK development is skewed heavily to industry,** whereas Germany, France, and the rest of Asia, Europe, and Online development is more balanced towards non-profits, universities, and community contributors.

as lmstudio-community, comfy, and mlx-community (together accounting for over 22% of recent downloads) focused on re-packing model formats for more efficient training and inference (e.g. quantization) becoming critical intermediaries. The USA's share fell to 15.8% in the recent period, while the "International/Online" category reached 23.8%, suggesting both geographic diversification and the continued rise of online power users as significant ecosystem participants. Because unaffiliated users and international online communities account for a large recent share, these national comparisons should be read as organization-headquarters attribution rather than a complete geopolitical accounting of model development labor.

## 3.1 The Rise of Communities & Unaffiliated Developers, over Industry Developers

**Model development and curation is shifting gradually away from large corporations and towards unaffiliated users and online community developers.** Perhaps most surprisingly, while industry used to occupy 70% of development before 2022, they now occupy only 37%. Most of this decline has been replaced by independent or unaffiliated users, having risen from 17% to 39% in 2025. However, the most significant upwards trajectory is from what Hugging Face classifies as *Community* developers, that may simply represent international/online volunteer teams, and often operate open source repositories. Since prior to 2025, their share of downloads has risen 1.5x. University and non-profit contributors have remained stable in their share over the last few years.

In Figure 4 we breakdown the share of downloads allocated to organizations of different types from each country. This analysis enables us to understand what types of organizations are associated with open model downloads by country. It is immediately clear that whereas US, Chinese, and UK downloads are dominated by industry, other European and Asian countries, as well as Online contributors, have a healthier distribution of non-profits, universities, and community contributors.

## 4 Shifts in Open Model Characteristics

The last five years of open models has seen nuanced shifts in the popularity of model architectures, modalities, training methods, documentation, and access. We collected supplementary information on these model attributes, beyond what is available in Hugging Face metadata, for the **Head Sample** (see Appendix E), and then ranked each model attribute by the largest relative shift from before January 2025 to after January 2025. Because the **Head Sample** is constructed from high-download models, these results characterize download-weighted shifts among prominent models rather than unweighted prevalence across the full Hub. In

| | ATTRIBUTE: CATEGORY | ≤2022 | 2023 | 2024 | ≥2025 | Δ% (<2025 \| ≥2025) |
|---|---|---|---|---|---|---|
| *Organizations* | Organization: Community | 0.0% | 7.9% | 3.8% | 9.6% | 1.7× |
| | Organization: User | 16.5% | 34.0% | 39.6% | 38.7% | 1.1× |
| | Organization: Non-profit | 4.7% | 4.1% | 6.4% | 5.4% | 1.0× |
| | Organization: University | 9.6% | 9.1% | 8.5% | 9.2% | 1.0× |
| | Organization: Company | 69.2% | 44.9% | 41.7% | 37.0% | 0.8× |
| *Model Access* | Data Availability: Not Disclosed | 9.8% | 23.4% | 31.6% | 43.1% | 1.7× |
| | Model Gating: Use Conditions + Share Info | 0.0% | 0.5% | 1.1% | 1.2% | 1.7× |
| | Data Availability: Disclosed & Unavailable | 2.1% | 1.2% | 2.2% | 2.9% | 1.6× |
| | License: Attribution | 0.2% | 0.6% | 3.3% | 3.0% | 1.6× |
| | Model Gating: Accept Conditions | 0.0% | 1.0% | 3.3% | 2.4% | 1.2× |
| | License: Undocumented | 17.3% | 29.5% | 27.7% | 31.2% | 1.1× |
| | License: Open Use | 81.4% | 54.1% | 58.3% | 55.1% | 0.9× |
| | License: Acceptable Usage Policy | 0.0% | 14.5% | 9.7% | 9.7% | 0.9× |
| | Data Availability: Disclosed & Available | 79.3% | 58.5% | 53.5% | 39.8% | 0.7× |
| *Model Modality* | Multimodal Generation | 0.0% | 0.8% | 1.3% | 3.3% | 3.4× |
| | Video Generation | 0.0% | 0.4% | 0.8% | 1.8% | 3.4× |
| | Text Generation | 21.0% | 11.0% | 12.5% | 20.3% | 1.7× |
| | Undocumented | 1.9% | 7.1% | 9.7% | 10.6% | 1.4× |
| | Tabular Models | 0.1% | 0.0% | 0.1% | 0.1% | 1.4× |
| | Audio Models | 1.0% | 7.4% | 9.0% | 9.2% | 1.2× |
| | Multimodal Embedding | 0.3% | 2.1% | 2.1% | 2.1% | 1.1× |
| | Image Generation | 0.3% | 27.9% | 25.0% | 23.0% | 1.0× |
| | Text Embed/Class | 75.2% | 28.5% | 26.9% | 22.9% | 0.7× |
| | Image Embedding | 0.2% | 14.8% | 12.5% | 6.6% | 0.5× |
| *Model Attributes* | Model Size | 215M | 827M | 1.76B | 20.8B | 17.0× |
| | Architecture: Mixture-of-Experts | 0.0% | 0.1% | 0.4% | 1.6% | 7.2× |
| | Methods: Quantization | 0.0% | 1.1% | 3.8% | 12.2% | 5.3× |
| | Methods: RLHF | 0.1% | 0.5% | 1.4% | 4.1% | 4.5× |
| | Architecture: Transformer: Text Decoder-only | 8.1% | 7.4% | 10.2% | 20.8% | 2.4× |
| | Methods: Instruction finetuning | 0.1% | 2.5% | 4.1% | 6.5% | 2.2× |
| | Architecture: Transformer: Speech | 0.8% | 3.8% | 6.9% | 7.7% | 1.4× |
| | Methods: Parameter-efficient finetuning | 0.7% | 6.1% | 6.5% | 8.8% | 1.4× |
| | Methods: Pretraining: Contrastive Learning | 2.1% | 5.8% | 11.7% | 10.2% | 1.2× |
| | Architecture: Diffusion-based Network | 0.1% | 20.0% | 18.9% | 19.5% | 1.1× |
| | Architecture: VAE / GAN | 0.0% | 5.1% | 2.6% | 2.8% | 0.9× |
| | Architecture: Transformer: Text Encoder-Decoder | 11.8% | 5.3% | 5.5% | 4.8% | 0.8× |
| | Architecture: CNN | 0.1% | 11.3% | 14.6% | 8.8% | 0.8× |
| | Architecture: Transformer: Text Encoder-only | 76.8% | 43.4% | 37.7% | 32.8% | 0.7× |
| | Architecture: LSTM / GRU | 0.4% | 0.2% | 0.2% | 0.2% | 0.7× |
| | Methods: Model Merging | 0.0% | 3.8% | 1.4% | 1.3% | 0.6× |

Table 1: **The incidence rate of model attributes over time, weighted by downloads in the Head Sample.** For each time segment, we measure the percent of downloads associated with models that have each ATTRIBUTE: CATEGORY characteristic. The Δ% (<2025 | ≥2025) column shows the multiplier change in a model attribute from before 2025 to during 2025. The table is intended as a descriptive effect-size summary: categories are filtered using model-level contingency tests over unique annotated models, not weekly download rows, and we emphasize the magnitude and direction of changes rather than the absolute $p$-values.

Table 1 we show the largest positive and negative shifts in the types of Model Developers, Model Access, Model Modality, and other Model Attributes. We devote this section to discussing key shifts in more detail.

## 4.1 Declining Training-Data Transparency

**Training-data transparency is on the decline. Model gating and restrictions are on the rise.** Table 1 shows a clear decline in both the availability, and disclosure of a models' training data. The Open Source Initiative defines open source AI models as those which have open model weights, but also "sufficiently detailed information about their [training] data."[6] Without sufficient training data disclosure, a released model is better described as "open weight" rather than fully "open source" under that definition. Whereas in 2022 over 79% of downloads were for models which disclosed their training data, in 2025 that value is only 39% (see Figure 9 for more details).

---

[6]https://opensource.org/ai/open-source-ai-definition

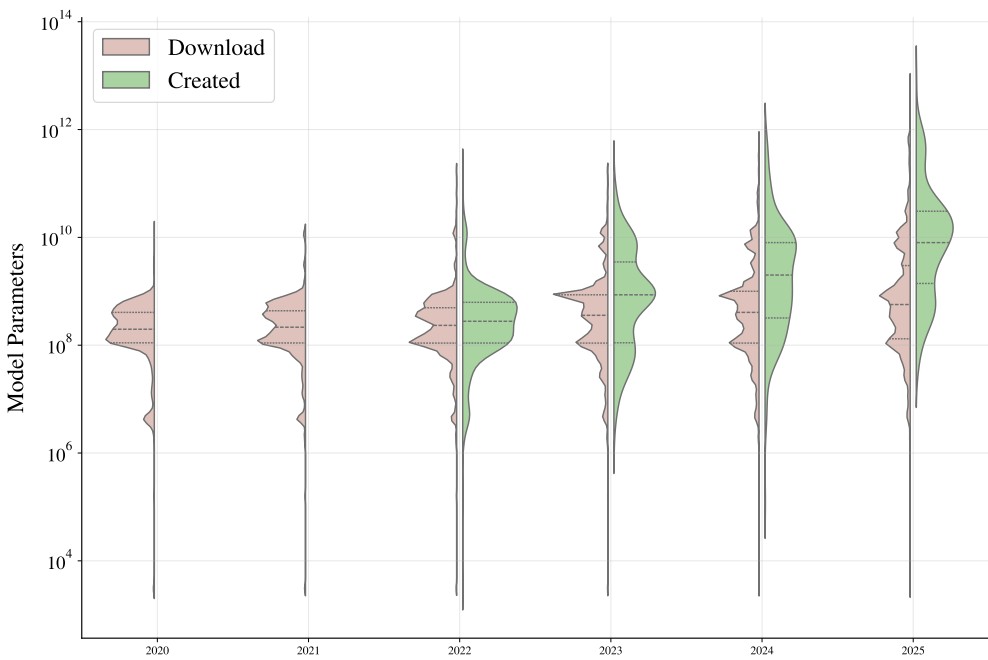

Figure 5: **The distribution of model sizes downloaded in each year (left, pink) and created in each year (right, green) is shifting over time.** The created statistics only begin in 2022, as Hugging Face did not record prior model creation times. We log-scale the downloads distribution prior to the violin plots' kernel density estimation to smooth and improve the visibility of the various model size peaks. The lines within each distribution represent the 25, 50, and 75 percentiles. We find the mean size of model download and creation are both rising, though the medians far less quickly as seen in Table 1.

Similarly, model access has become more restrictive, both in terms of gating models (requiring users to accept conditions or share their information first), and more restrictive, tailored, licensing. Many major models, such as Meta's Llama series, are now gated, comprising over 3.6% of all model downloads. As for model licenses, a smaller portion of downloads are allocated to models that provide any license information, as compared to prior years. And when licenses are documented, Open Use licenses are on the steep decline, whereas non-commercial, or attribution requirements are on the rise (see Figure 10 for more details).

## 4.2 Rising Model Sizes

**The average size of a downloaded model is increasing rapidly, with rising compute availability, advances in quantization, and mixture-of-experts architectures.** In Figure 5 we see the distribution of model sizes according to their download and creation prevalence over time. First, the mean size of a downloaded model has increased from 827M in 2023 to 20B in 2025. Note that the median size of a downloaded model remains much lower: 326M in 2023, and 406M in 2025. This suggests that most of the community's compute affordances have risen more modestly, whereas some power users have scaled to multi-billion parameter architectures. Much of this power scaling appears to be enabled by a few innovations: aggressive quantization for inference-time memory savings, and mixture-of-expert architectures that have massive parameter counts but many fewer that are active at a time. For instance, Kimi K2 (Team et al., 2025) is a popular 1T parameter mixture-of-expert model, with only 32B active per token. Other popular hundred-billion parameter mixture-of-experts examples include the DeepSeek-V3, Mixtral, Grok-1, MiniMax, and Snowflake Arctic series. Comparably, Nvidia's Nemotron series comprise fully dense networks in the hundreds of billions of parameters. The growing adoption to these massive models demonstrates a continued reliance on scaling to achieve performance.

**The mean size of created models consistently outpaces the mean size of downloaded models, suggesting developers are investing in larger models before most deployers have inference capacity to use them.** Figure 5 shows not only that the mean size of created models is larger than the mean size of downloaded models, but that it is rising much more quickly. This clearly suggests developers are

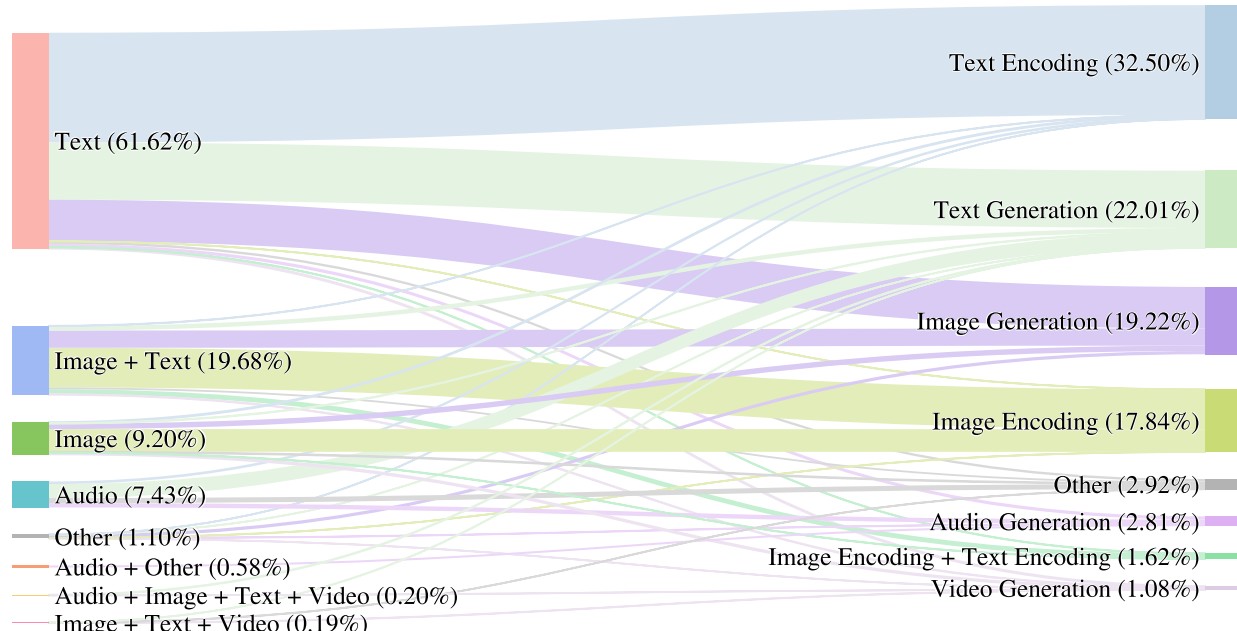

Figure 6: We illustrate the **Head Sample**'s aggregate model modality distributions, weighted by allocation of downloads. The model input modalities are depicted on the left, and the model output modalities on the right. We find model inputs are predominantly text (61%), followed by text and images together (20%), then images (9%), audio (7%), followed by other combinations. The outputs are ranked as text encoding (33%), text generation (22%), image generation (19%), image encodings (18%), audio generation (3%), and video generation (1%). **We find the output modalities are more heterogeneously distributed than that input modalities on aggregate.**

scaling faster than most deployers can reasonably adopt. The time lag between development and deployment may span over a year.

### 4.3 The Shift to Generative Multimodal Architectures

**The ecosystem is quickly migrating towards multimodal, generative systems, and away from encoder-based discriminative architectures.** Table 1 shows two clear temporal trends in the model modality and architecture shifts. First, we see a shift in model architectures. Encoder-based models, typically used to produce embeddings for retrieval or classification tasks are declining in favor of decoder-based generative models. Similarly, convolutional neural networks (CNNs), long-short-term-memory cells (LSTMs), and generative-adversarial networks (GANs) are falling off in popularity. Second, there is a striking rise in the adoption of mixture-of-experts architectures, and for a greater variety of multimodal outputs, such as 3.4x for video. Increasingly, models incorporate combinations of text, images, audio, and video generation in their output capabilities. This emphasis on output capabilities even over input capabilities is illustrated in Figure 6, where output modality heterogeneity is higher than input modalities.

**Training and inference efficiency methods are becoming an essential stage of model development.** As deployers scale to larger models, finetuning-adaptation and inference efficiency become paramount to support these systems. Unsurprisingly, the incidence of parameter-efficient finetuning (PeFT) (Houlsby et al., 2019; Pfeiffer et al., 2020) and quantization (Nagel et al., 2021; Lin et al., 2024) have increased by 40% and 500% respectively since prior to 2025. While model merging (Yang et al., 2024a) and knowledge distillation (Hinton et al., 2015; Liang et al., 2021) methods have not risen in popularity, reinforcement learning and instruction tuning (RLHF) (Ouyang et al., 2022), which often leverage larger "teacher" models, have become significantly more common-place in terms of usage since 2024. This suggests a shift from certain techniques leveraging teacher models to other techniques leveraging reward models or synthetic data generators.

# 5 Discussion

Our longitudinal analysis of 2.2B model downloads across 851k models exposes a rapidly evolving open AI ecosystem; one that is simultaneously decentralizing and consolidating, expanding in technical capacity while narrowing in transparency. These download-based trends reveal important structural tensions in how open AI develops, who participates, and which actors come to exercise influence over global model distribution, underscoring the need for continuous ecosystem monitoring.

**Rebalancing of Usage Share and the Cycles of Decentralization.** Between 2021 and 2024, the open model ecosystem underwent a marked diffusion of download-based usage share. The download share of the three dominant U.S. industry organizations, Google, Meta, and OpenAI, declined sharply, falling from peaks of >40–60% to a marginal position by 2025. Unaffiliated developers and loosely coordinated online communities became the primary drivers of model development, in some periods accounting for more than half of all downloads. This shift coincided with the explosive adoption of diffusion-based generative models and LoRA-based customization workflows, which dramatically lowered the barrier to participation.

However, 2025 shows signs of recentralization in download-based usage, driven not by U.S. incumbents but by rising Chinese developers. DeepSeek and Qwen together captured 14% of all downloads in the most recent year, and China's attributed national share rose to 17.1%, surpassing the United States (15.7%) for the first time in our organization-headquarters attribution. This pattern of diffusion following a technological shock, then consolidation around leaders of the next technological wave suggests a cyclical rather than linear evolution of the ecosystem.

**The Rise of Intermediary Developers.** A defining structural transformation is the emergence of a new intermediary layer: organizations specializing not in training base models, but in re-packaging, quantizing, adapting, and refactoring them for community use. Groups such as lmstudio-community, comfy, and mlx-community now represent over 22% of downloads in the most recent year. These intermediaries play an infrastructural role analogous to cloud providers in traditional computing: they translate cutting-edge frontier models into practically deployable artifacts.

This development reflects the industrialization of open model reuse. As model sizes increase on average from 827M parameters in 2023 to 20B in 2025, efficient inference and adaptation become essential. Intermediaries specialize in this efficiency layer, shaping not only which models become accessible to typical users, but also how innovations diffuse across the ecosystem.

**Technical Transformation and Shifting Norms.** The ecosystem's technical profile has shifted toward larger, more multimodal, and more computationally efficient architectures. The incidence of video generation and multimodal generation models grew 3.4×, while mixture-of-experts architectures increased 7.2×. Meanwhile, quantization techniques surged 5×, and parameter-efficient finetuning rose by over 40%, indicating their centrality to contemporary deployment practice.

Importantly, the rise in scale is concentrated in the long-tail of high-capacity deployers: while the mean downloaded model grew to 20B parameters, the median rose only modestly from 326M (2023) to 406M (2025). This divergence indicates expanding inequality between power users capable of hosting multi-billion-parameter models and typical developers who remain limited by compute constraints.

**Declining Transparency.** Despite rapid growth in model availability, training-data transparency has deteriorated substantially. Models disclosing available training data fell from 79.3% (2022) to 39% (2025), and for the first time, open-weight models lacking sufficient training-data disclosure account for more downloads than models satisfying core open-source AI documentation criteria. At the same time, model gating, attribution requirements, and use-condition licenses all increased.

This shift reflects real tensions: as models grow larger and more commercially valuable, incentives to restrict access intensify. Yet the decline in transparency raises profound questions for reproducibility, governance, and accountability; precisely in the period when open models are absorbing heightened geopolitical and economic relevance.

## Ethics Statement

**Limitations.** It is important to note that Hugging Face download metrics, or any other metrics, are at best proxies for real usage. We make the case in Section 2 for why we believe this data, with sufficient de-duplication and filters, provides the most accurate estimate of model usage, though it still remains imperfect. As a result, these metrics best serve the analysis of broader trends and relative popularity over time, rather than precise numerical designations of deployment, revenue, or economic concentration. We explicitly caution against interpreting national or developer download shares as direct measures of geopolitical strength, commercial value, or downstream social impact. The longitudinal record also excludes models that were removed, made private, or otherwise unavailable at the time of data collection. Further, some time-varying metadata such as licenses, gating, and model cards are observed from current public metadata and may not perfectly reconstruct the historical state of a model at the time a download occurred.

**Privacy.** This study analyzes aggregate download and metadata statistics from the publicly available Hugging Face Model Hub to understand structural shifts in the open AI ecosystem. All data used are non-personal and pertain exclusively to publicly released AI models, their metadata, and anonymized download counts. No individual user identifiers or personal information were accessed or retained. We decided to only identify organizations' countries, not for individual users, to retain anonymity.

**Artifact Release.** By releasing the public dashboard and annotation methodology openly, we aim to enable reproducibility, critical analysis, and further community research into usage concentration, openness, and data transparency. The data and code are shared under an open license consistent with responsible research and privacy guidelines.

**Responsible and fairly compensated data labor.** We recognize the importance of fair compensation and protections for data laborers in AI Du & Okolo (2025). Annotators for this project were employed through Upwork and received $25/hour–well above average U.S. minimum wage U.S. Department of Labor (2025). Strict timelines were not imposed and annotators were able to determine and work according to their own schedules. Additionally, we value the technical expertise and diverse perspectives our annotators contributed and continuously improved and expanded our taxonomy with their input. We believe transparency regarding data labor practices is essential and all annotators have accepted the offer to be recognized for their work in the acknowledgments.

**Framing.** Finally, we acknowledge potential ethical risks associated with framing AI development through geopolitical or corporate competition. Our analysis seeks to document ecosystem dynamics, not to reinforce nationalistic narratives or proprietary advantage. Country labels are based on organization headquarters, while unaffiliated users are intentionally not assigned countries for privacy reasons; this design preserves privacy but limits geopolitical interpretation. We encourage subsequent work to combine quantitative analysis with qualitative inquiry into the social and cultural dimensions of open AI ecosystems.

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

# Appendix

# A  Related Work

**Societal and Political Analyses of AI.** The discourse surrounding the AI ecosystem is modulated by dichotomies that obscure deeper power dynamics. Much work has been done in the field of critical algorithm studies to understand the societal and political currents steering AI development and deployment. Particularly relevant to this work are; Lehdonvirta et al. (2024b)'s documentation of a "Compute North vs. Compute South" divide, where computational resources concentrate in wealthy nations and reinforce extractive infrastructures. Crawford (2021b) and Noble (2018) identify how social hierarchies embedded in technical systems operate under the guise of neutrality while perpetuating existing power structures. The dichotomy between closed AI and open AI reveals similar contradictions: despite open models costing 6x less to deploy.

**Economic and Concentration Analyses of AI.** Other work on documenting open model characteristics find that closed models dominate 80% of usage and 95% of revenue, exposing how "openness" fails to translate into meaningful access (Nagle & Yue, 2025). These divides intersect with institutional shifts documented by (Ahmed et al., 2023), who show 96% of recent AI advances originate from industry, though Vipra & West (2023) argue the fundamental barrier remains computational access rather than institutional affiliation. Together, these overlapping binaries fragment understanding while power concentrates across multiple dimensions simultaneously. These divisions mask how advantage compounds through control of compute, data, and talent markets. Building on these dichotomies, the open/closed binary might suggest that open model ecosystems represent a democratic alternative to concentrated power in closed systems, yet empirical studies reveal a more complex reality.

**Geopolitical Analyses of AI.** Work documenting these ecosystems includes Lambert's analyses of China's open-source trajectory and model adoption patterns (Lambert, 2025b;a), a NIST evaluation of the Chinese DeepSeek models (for AI Standards et al., 2025), and Epoch AI's study of open models (Cottier et al., 2024). These empirical platform studies demonstrate that openness alone does not distribute power: computational resources, data access, and technical expertise create hierarchies within open ecosystems, highlighting the importance of studying said ecosystems.

**Hugging Face Ecosystem Analyses.** Finally, a growing body of research uses Hugging Face as a testbed for studying open model ecosystems at scale. Ecosystem analyses of the Hugging Face Hub examine the landscape of over 2 million models (Laufer et al., 2025b; Horwitz et al., 2025a) and analyze community dynamics (Ait et al., 2023). Research on model evolution tracks maintenance patterns and versioning challenges (Castaño et al., 2024; Ajibode et al., 2025), while studies of reuse practices reveal naming conventions, defects, and ecosystem dependencies (Jiang et al., 2023; 2025; Yang et al., 2024b). Recent data-management work frames models, datasets, licenses, metadata, and provenance as ML assets requiring curation, discovery, utilization, lineage, and unified indexing infrastructure (Wang et al., 2025). At large, systematic reviews synthesize existing Hugging Face research and validate the platform's suitability as an empirical research site (Jones et al., 2024; Ait et al., 2025). These studies collectively demonstrate Hugging Face's emergence as critical infrastructure both for hosting the open model ecosystem and for understanding its development dynamics.

# B  Data Sources

We summarize our data collection in Table 2. These data sources are collected from a mix of internal Hugging Face download logs, public model metadata on the Hugging Face hub, as well as manually or automatically inferred model metadata.

## B.1  Hugging Face Model Download Data

The data on model downloads is publicly available on Hugging Face for the past 30 days for each model. To access download data for all models across all time, we leverage internal databases to collect information on download statistics, aggregated on the weekly level. We filter data to only cover models that are *currently* publicly available, i.e., at the time of data collection, using the Hugging Face API `list_models` function.

| Data | Src | Cov | Description |
|---|---|---|---|
| **Usage Logs** | 🤗 | All | Weekly downloads for public models from June 2020 to July 2025. |
| **Model Size** | 🤗🤖 | All | Estimated model parameters, using both safetensors and regressing model file byte size against the parameter counts. |
| **Model Developer** | 🤗✏️ | All | The model developer, and their organization's headquarters (including international and online). |
| **Architecture & Modalities** | ✏️ | Head | The model's architecture, and its input and output modalities. |
| **Training Methods** | ✏️ | Head | The methods used to train this model, or derive it from another model |
| **Language(s)** | 🤗✏️ | All | The languages used to train this model, if text-based. |
| **Model & Data Access** | ✏️ | Head | If the model is gated, and if the training datasets are documented and accessible. |
| **Model Graph** | 🤗✏️ | Head | A graph showing what model(s) each model was derived from. |

Table 2: **A list of the data collected, their sources, and coverage.** We list the source(s), whether from Hugging Face APIs (🤗), automatic crawling (🤖), or manual collection (✏️), and whether the metadata covers *All* models, or just the *Head* sample.

This creates a survivorship limitation: models that were public historically but later removed, renamed, or made private are not included in the reconstructed historical record.

### B.2  Hugging Face Hub Model Metadata

The metadata fields that augment each row of the dataset have multiple sources, though most are returned by specific API endpoints of the Hugging Face Hub. The organization metadata is scraped from the Hugging Face website itself since detailed information is not yet available for structured querying. The model cards data column is extracted by a dataset maintained and updated daily by Hugging Face ML librarian Daniel van Strien (https://huggingface.co/datasets/librarian-bots/model_cards_with_metadata). The modalities are automatically extracted using heuristics based on structured data extracted from the Hugging Face documentation and tasks page (https://huggingface.co/tasks). A more detailed and extensive overview is included in the dataset cards of each of the released datasets. Unless otherwise noted, metadata reflects the public model state available during our collection process; historical license, gating, model-card, and data-disclosure changes may therefore be imperfectly reconstructed.

### B.3  Geographical Metadata for Model Developers

In our annotations for national association, we only focus on organizations, not user accounts, to preserve privacy of individual contributors. For each model developer, the authors manually annotated the country associated with the organization. This is done by finding evidence of the organization's website on the Hugging Face page. The country is chosen as the organizations' headquarters. For certain organizations that are primarily based online, or exist without a primary headquarters based in one nation, we label them as "Online", or "International", respectively. For individual contributors to the platform, where no organization appears directly affiliated with the release of the model, we label the origin as "Individual". This attribution intentionally avoids inferring individual users' countries, but it also cannot represent multinational development teams, globally distributed companies, or repackaged uploads whose uploader is not the original developer.

## C  Detailed Results

In Figure 9 we show fine-grained temporal trends for model architecture (top), model development methods (middle), and data disclosure/transparency (bottom). Additionally, Figure 10 shows how the proportion of license types has changed over time. Lastly, Figure 11 shows the download-weighted graph of models that have been derived and in what ways they have been derived. These figures provide greater details into the temporal trends and model derivations on the Hugging Face hub, as weighted by usage metrics.

## D  Rolling Window Sensitivity

Our main temporal concentration figures use a one-year Rolling Window Filter to reduce the influence of legacy automated downloads. Figures 7 and 8 compare this choice with no recency cap and with a stricter

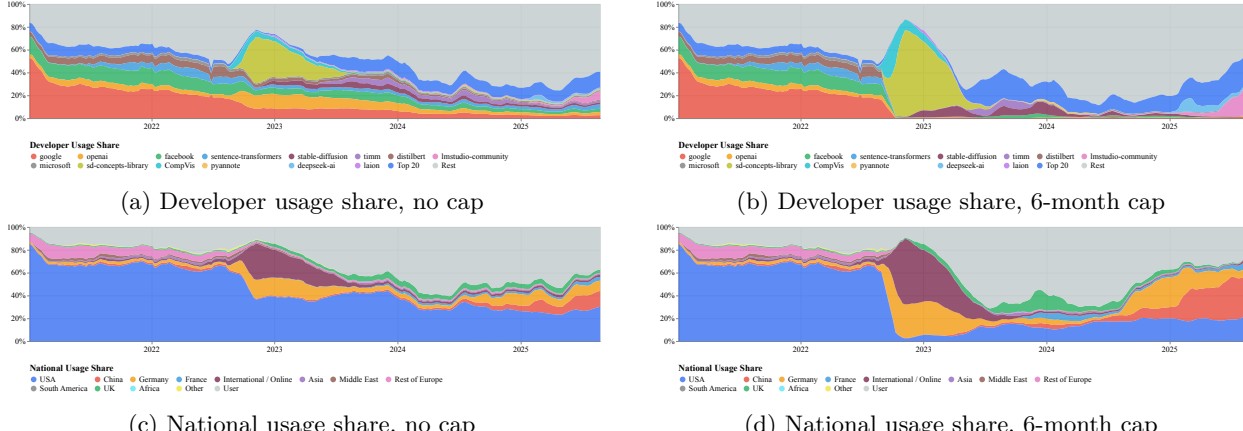

Figure 7: Sensitivity of temporal usage-share rankings to the Rolling Window Filter parameter. The main paper uses a one-year cap; no cap overweights older automated dependency downloads, while a six-month cap preserves the direction of the central trends with sharper recent shifts.

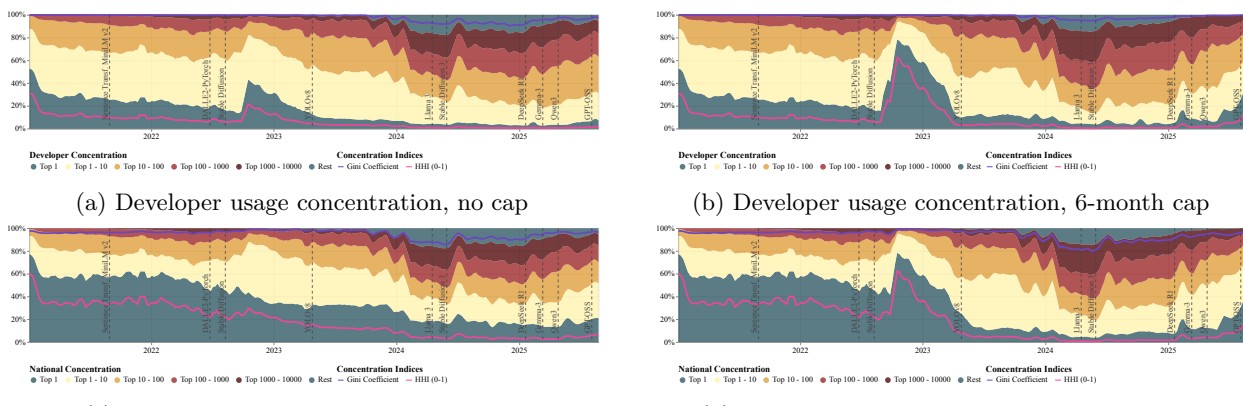

Figure 8: Sensitivity of usage concentration estimates to the Rolling Window Filter parameter. The renewed 2025 increase in concentration is visible under both the main one-year window and a stricter six-month window; removing the cap makes legacy automated downloads more prominent and therefore less suitable for measuring contemporary usage.

six-month cap. Without a cap, older BERT-, CLIP-, and embedding-era models remain disproportionately visible in later years, consistent with automated dependency pulls rather than contemporary model choice. Under the stricter six-month cap, the same qualitative pattern remains but is sharper: early concentration declines through the diffusion period and rises again in 2025. We therefore retain one year as a conservative middle ground that preserves sustained adoption while limiting stale automated downloads.

# E   Annotation Taxonomy & Design

To gain meaningful insight into the specific features and types of models that populate the Hugging Face Ecosystem, we needed to collect information beyond what is available programmatically or documented within model cards. We recruited four annotators to perform this manual analysis: two based in the United States, one in Serbia, and one in Egypt. All annotators had a strong background in computer science/machine learning and completed multiple rounds of training, guided by expert annotators, before beginning any real annotation tasks.

Annotations were produced with detailed written guidelines, regular review meetings, and adjudication of uncertain cases through discussion and taxonomy updates before downstream analysis. After annotation, an expert NLP researcher reviewed a random sample of approximately 10% of all annotator labels. Across all

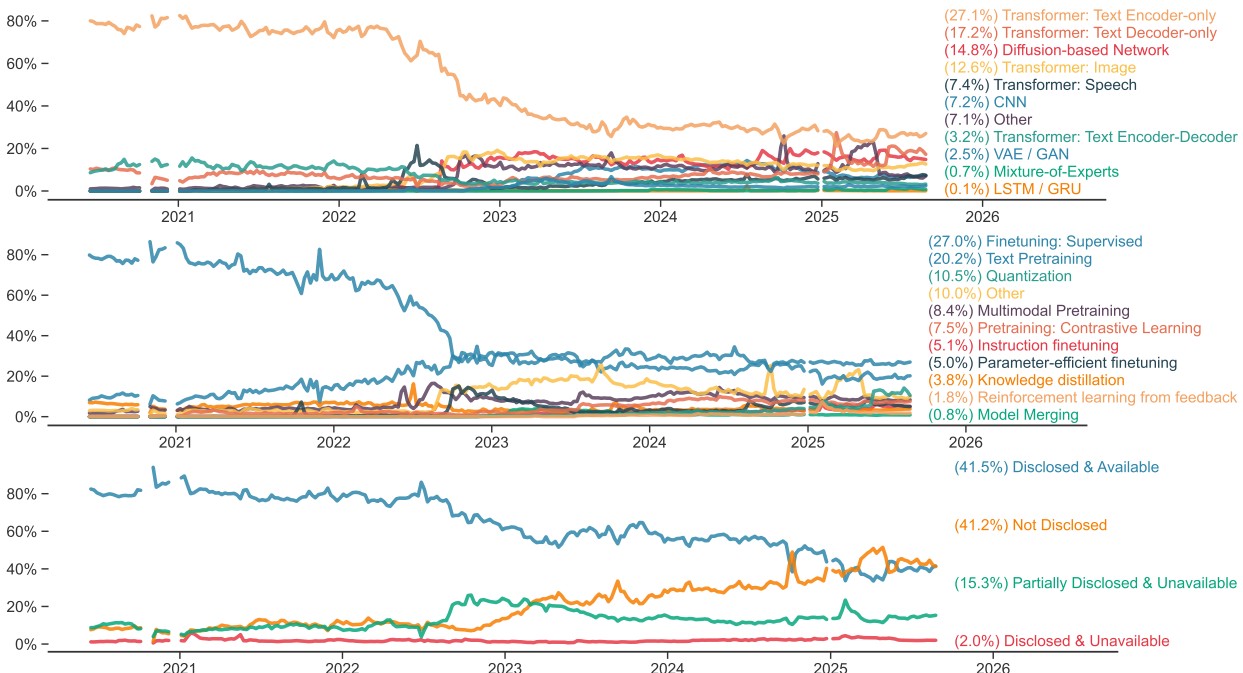

Figure 9: These plots measure the portion of downloads associated with different model attributes. **Top:** Model architectures over time. **Middle:** Training and inference methods over time. **Bottom:** The disclosure and availability of a models' training data.

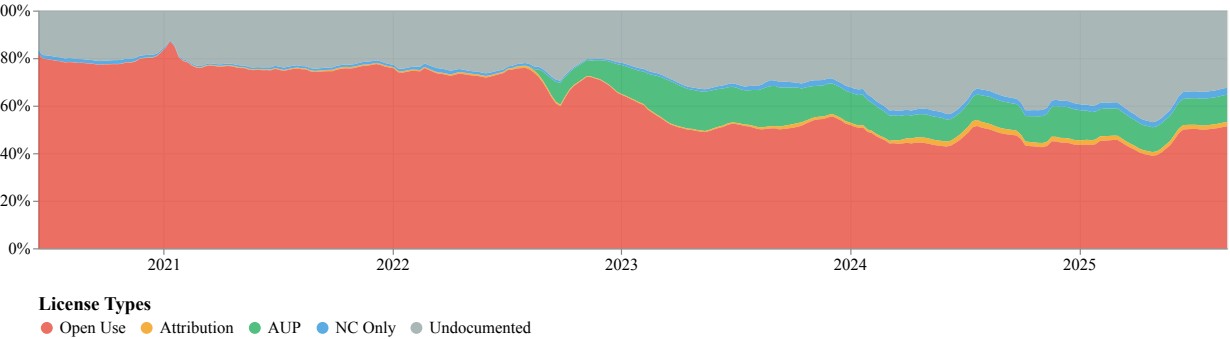

Figure 10: The portion of downloads associated with models of a given license type. **Open Use licenses are on the decline, mainly replaced by licenses with Acceptable Usage Policies (AUPs).**

annotation fields, reviewed labels were at least 90% accurate; model architecture had the lowest agreement but still exceeded this threshold. This expert review gives us confidence in the annotation quality, and we report it as a sampled expert-review estimate rather than a separately benchmarked model-style confidence interval.

When necessary, we instructed annotators to search for and consult additional sources (e.g., the accompanying publication or associated GitHub repository) in order to answer the following questions about each model:

- Is the model gated on Hugging Face?

- What are the model input and output modalities?

- What languages are the models trained to cover?

- Is the model derived from another model?

- What is the model architecture?

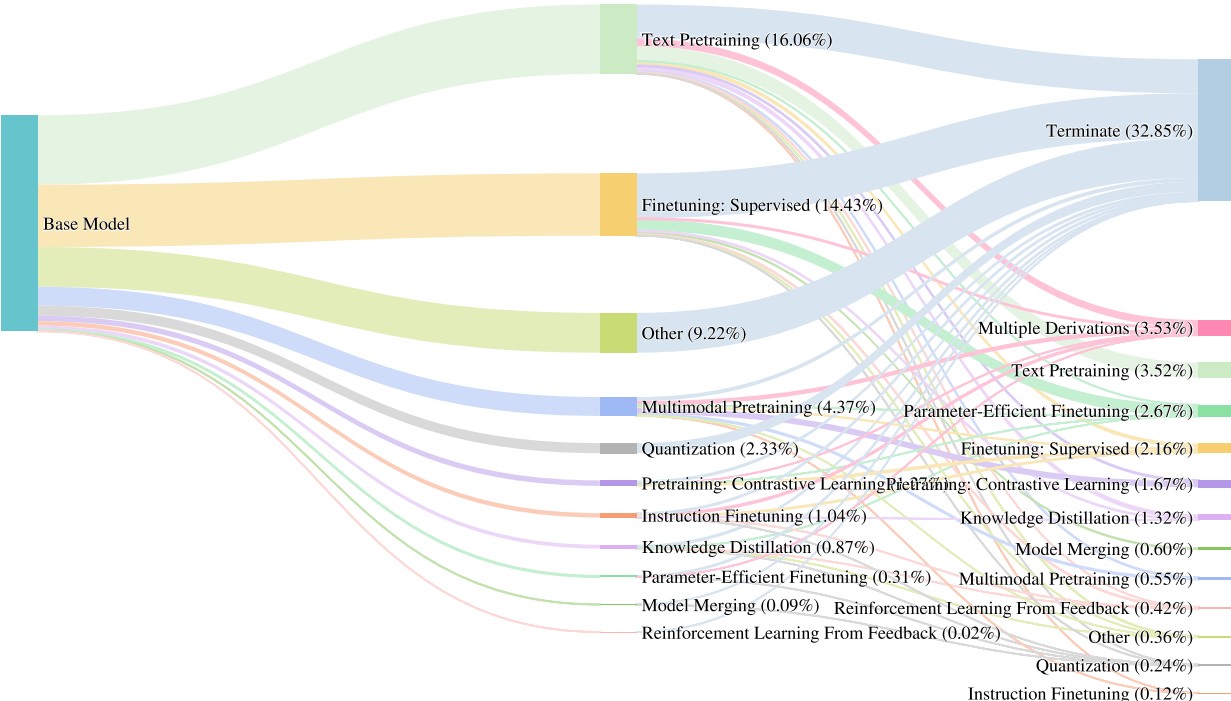

Figure 11: We illustrate a network derivation graph, showing the portion of downloads allocated to models derived according to different methods. Each column sums to 100%. We find that supervised finetuning is the most popular type of derivation (21%), followed by continued pretraining (16%), and quantization (4%). Models that undergo a second round of derivations see additional pretraining (7%), parameter-efficient finetuning (5%), and finetuning (4%). And a full 7% of model downloads are allocated to systems with 2+ series of derivations.

- What training methods were used?

- Are the datasets used to train the model fully documented and publicly available?

### E.1 Model Gating

Model authors have the ability to gate access to their models on Hugging Face. Often, users gain access by simply agreeing to share their username and email address with the model author; however, some models require additional information or agreement to specific terms of use. To document which models are open access and which are gated, we have provided the following instructions to our annotators:

---

**Task: Is the model gated on Hugging Face?**

**Description**

On Hugging Face, some models can only be accessed or downloaded once some terms have been accepted or information entered. This might be as simple as clicking "Agree and Access Repository" on the main page, or filling out additional information. Please indicate if this is the case.

**Multi-label Options**

- N/A

- Accept to share username & email

- Other agreement/info requirements

---

### E.2 Modality

For each model, we note the associated input and output modalities. For example, generative models may support a single modality (e.g., text → text) or multimodal (e.g., text → image), while other models are trained for specific tasks (e.g., text → classification). This allows us to understand how the priorities of model authors have shifted across modality. We have provided the following instructions to our annotators to document input and output modality:

---

**Task: What is the input and output modality of the model?**

**Description**

Identify the data types that the model takes as input and produces as output. There will be two separate columns for this task, one for documenting input modality and one for documenting output.

- Models may be a single modality (e.g., text → text) or multimodal (e.g., image → text).

- If the model is not generative, list the output type (e.g., text → classification).

- Refer to the model card, and, if available, demo and associated paper.

- Use the predefined labels for consistency.

- If the model uses a modality not covered by the provided options, select `Other` and clarify in the Note column.

- Write the exact label as formatted in the multi-label options below to ensure the annotations are machine-readable.

- If you specify `Other` use the Note column to explain.

---

**Multi-label Input Options**

- Text

- Image

- Video

- Speech

- Tabular

- Other

**Multi-label Output Options**

- Text Generation

- Image Generation

- Video Generation

- Speech Generation

- Tabular

- Text Classification

- Text Sequence Classification

- Image Classification

- Image Segmentation

- Image Bounding Boxes

- Text Embedding

- Image Embedding

- Other

### E.3 Languages

In an effort to measure linguistic inclusion and gain insight into the rate at which language groups are represented across popular Hugging Face models, we document each instance a model is trained or finetuned on a specific language dataset. In addition to natural language, we also note cases where a model is trained on coding languages. We have provided the following instructions to our annotators to document language:

**Task: What languages are the model trained to accommodate?**

**Description**

Determine which natural and coding languages the model is designed to handle. This typically refers to the language(s) present in the training data for text or speech models.

- If language information is unavailable on the model card, check the associated paper, dataset links, or GitHub documentation.

- Use ISO language codes (e.g., `EN` for English, `DE` for German) where possible.

- If multiple languages are supported, document them all in a comma-separated list.

- If the model does not generate language but, for example, has pre-defined classification labels, list the language(s) of those labels.

- If there are clearly over 50 languages, then simply mark it as `multilingual`.

- If this model was derived from another, then only include the languages used in the derivation process (e.g., fine-tuning), not the base model's pretraining languages.

## E.4   Model Derivation

A model is considered derived if it inherits the weights of a pre-existing base model before conducting additional training, quantization, or other adaptations. To gain insight into upstream influence and which models power the Hugging Face ecosystem, we have provided the following instructions to our annotators to document derivations:

**Task: Is the model derived from another model?**

**Description**

Indicate whether the model is derived from another pre-existing model. A model is considered derived if it takes another model's weights and conducts additional training, quantization, or model merging. A model is not considered derivative if it uses the same architecture or code.

- This may be mentioned in the model card (often under `base model`), the repository, or the paper.

- Provide the full name of the base model if known.

- If the model was trained entirely from scratch, indicate `No`.

- Input the exact Hugging Face model name, so it can be looked up automatically. For example, for BERT it would likely be `google-bert/bert-base-uncased`.

## E.5   Architecture

The architecture of a model determines how it reasons and processes input. Documenting which architectural choices model authors have implemented across time allows us to identify dominant paradigms (e.g., transformers or diffusion models) and assess how shifts in architecture reflect emerging tasks, modalities, or compute constraints. We have provided the following instructions to our annotators to document model architectures:

**Task: What is the model architecture?**

**Description**

Specify the architecture used by the model, such as `Transformer`, `ResNet`, `UNet`, `LSTM`, etc.

- Use precise architecture names as described in the model documentation or paper.

- Include whether the model is a variant or combination (e.g., Vision Transformer + Decoder).

- If the model encodes multiple modalities, comma-separate the labels.
  For example: `Transformer: Text Encoder-only, Transformer: Image Encoder-only`.

- A model might also combine architectures, such as being both a `Transformer` and a `Mixture-of-Experts` model.

- Common examples have been pre-categorized here.

**Multi-label Options**

- Transformer: Text Encoder-only

- Transformer: Text Decoder-only

- Transformer: Text Encoder-Decoder

- Transformer: Image Encoder-only

- Transformer: Image Decoder-only

- Transformer: Image Encoder-Decoder

- Transformer: Speech Encoder-only

- Transformer: Speech Decoder-only

- Transformer: Speech Encoder-Decoder

- Transformer: Unknown

- LSTM

- GRU

- CNN

- Diffusion-based Network

- Variational Autoencoder

- Mixture-of-Experts

- Generative Adversarial Network (GAN)

- Unsure

- Other

## E.6   Training Methods

The training methods used when creating a model impact how the model performs on specific tasks. Documenting which training techniques model authors have opted for over time allows us to trace the adoption of popular new techniques (e.g., fine-tuning, reinforcement learning and quantization), and assess how the field balances the reuse of existing models with the creation of new ones. We have provided the following instructions to our annotators to document model training methods:

**Task: What training methods were used?**

**Description**

Describe the training strategy applied to the model. This could include pretraining methods (e.g., masked language modeling, contrastive learning), fine-tuning approaches, reinforcement learning (e.g., RLHF), or other techniques (e.g., instruction tuning, distillation).

- Refer to the model card or original paper.

- We propose concise and standard terminology below. If you list `Other`, then please also suggest what it is.

- If this model was derived from another, then only include the training methods used to make the derivation (e.g., fine-tuning, not the base model's pretraining).

**Multi-label Options**

- Pretraining: Masked Language Modeling (MLM)

- Pretraining: Causal Language Modeling (CLM)

- Pretraining: Denoising Autoencoder

- Pretraining: Contrastive Learning

- Pretraining: Variational Autoencoder

- Pretraining: Multimodal joint-embeddings

- Pretraining: Supervised

- Pretraining: auto-regressive image generation

- Pretraining: Next Sentence Prediction (NSP)

- Finetuning: Supervised

- Finetuning: Prompt-based tuning

- Parameter-efficient finetuning

- Knowledge distillation

- Instruction finetuning

- Multi-task finetuning

- Reinforcement learning from feedback

- Adversarial Training

- Quantization

- Model Merging

- Unsure

- Undisclosed

- Other

### E.7 Training Data Availability

An essential component of transparent model development is data provenance. In order to monitor trends in openness, we document whether training datasets are publicly available or proprietary. We have provided the following instructions to our annotators to document training data availability:

---

**Task: Are the datasets used to train the model disclosed and publicly available?**

**Description**

Find out if the Hugging Face model card or associated paper for the model disclosed what training datasets were used. We would also like to know if those datasets are publicly available or proprietary. If this model was derived from another, then we only care about the training methods used to make the derivation (eg. fine-tuning datasets but not the base model's pre-training datasets).

- **Not disclosed** — There is no information on the datasets used at all.

- **Partially disclosed: unavailable** — General information is provided (e.g., `web data`) but nothing else.

- **Disclosed: unavailable** — A detailed list of the dataset sources is provided, but they are proprietary.

- **Disclosed: available** — The dataset(s) are disclosed, named, and public.
  In this case, please add a note on where you found the dataset(s) listed.
  All of the main datasets must be publicly available for it to count as `Disclosed: available`.

- If the model is derived, only consider the data used for the derivative.
  For example, if the model is a derivative of `google-bert/bert-base-uncased` (which has publicly disclosed and available training data) but does not mention any additional data, label it as **Not disclosed**.

**Label Options**
- Not disclosed

- Partially disclosed: unavailable

- Disclosed: unavailable

- Disclosed: available

---

