# OpenReview forum: "Economies of Open Intelligence: Tracing Power & Participation in the Model Ecosystem"
_TMLR — Decision pending for TMLR_

### Review · Reviewer_ob6G · 2026-04-14

**Summary Of Contributions:**

**Summary**

This paper presents a large-scale longitudinal analysis of the Hugging Face model ecosystem (2020–2025), combining download data with extensive metadata to study how influence, participation, and model characteristics evolve over time. Using economic concentration metrics (e.g., HHI, Gini), the authors analyze trends across models, developers, and countries and report shifts such as decreasing US industry dominance, rising contributions from the community and China, increasing model scale and multimodality, and declining transparency. The paper also releases the dataset and an interactive dashboard for ecosystem monitoring.

**Key strengths:**
* Comprehensive, large-scale dataset covering the majority of Hugging Face downloads over multiple years.
* Timely and policy-relevant analysis of the open model ecosystem.
* Clear and interpretable use of economic metrics to study concentration dynamics.
* Public release of dataset and dashboard as a valuable resource for future research.

**Key weaknesses:**
* Downloads as a proxy for usage and influence. The analysis equates (filtered) download counts with model usage and even “economic power,” but downloads may not reflect real deployment, repeated usage, or API-based access. While the authors acknowledge this limitation, some conclusions are phrased strongly and partially overstate what this proxy can support.
* Country attribution is potentially misleading. Models are assigned to countries based on developer organizations, but large companies (e.g., Google, Meta) operate globally. This can distort the geopolitical analysis, as models may be attributed to the US even if they were developed elsewhere.
* The download counting heuristic is not well justified. Counting at most one download per user per day introduces inconsistencies (e.g., two downloads on adjacent days count as two, while two on the same day count as one). It remains unclear why this definition best approximates “usage,” and alternative strategies (e.g., stronger deduplication) should be discussed.
* Rolling window filter biases usage estimates. The assumption that models are replaced within one year conflicts with the paper’s own observation that adoption often lags behind development. In practice, older models (e.g., Llama-2) remain widely used, especially in academia. The filter, therefore, likely underestimates sustained usage and biases the analysis toward more recent models.
* Missing discussion of alternative usage signals. The paper focuses exclusively on download data, but other proxies—such as mentions of models in academic publications—could provide complementary perspectives on adoption, particularly in research settings. Discussing such alternatives would strengthen the overall conclusions.

**Audience:**

Yes

**Audience Explanation:**

The paper addresses the structure and evolution of the open model ecosystem, which is highly relevant to researchers working on large language models, open-source AI, and AI governance. Its large-scale empirical analysis and released dataset are likely to be of interest to a broad subset of the TMLR audience, particularly those studying model development, deployment trends, and ecosystem dynamics.

**Broader Impact Concerns:**

The paper analyzes aggregate, publicly available data from the Hugging Face ecosystem and does not involve personal or sensitive information. The authors also include an ethics statement discussing limitations, privacy considerations, and potential risks in framing geopolitical dynamics, which is appropriate for this work.

One minor suggestion is to further emphasize the risk of overinterpreting proxy-based measures (e.g., downloads as “power”) in policy or geopolitical contexts, as this could lead to misleading conclusions if taken at face value. Overall, however, the ethical implications are adequately addressed.

**Claims And Evidence:**

Yes

**Claims Explanation:**

The paper provides extensive empirical evidence based on a large-scale longitudinal dataset, and the analysis is generally clear and well-supported for descriptive trends within the Hugging Face ecosystem. The use of economic concentration metrics and detailed metadata strengthens the credibility of the reported patterns.

However, the central claims rely on proxy measures, most notably download counts as a stand-in for usage and “economic power.” In addition, methodological choices such as the rolling window filter, download deduplication strategy, and country attribution introduce potential biases that are not fully validated. As a result, while the evidence convincingly supports the observed trends in the dataset, some higher-level interpretations and conclusions are somewhat stronger than what can be strictly justified by the underlying measurements.

**Requested Changes:**

**Critical:**
* *Better qualify claims based on download data*.
The paper should more clearly distinguish between downloads, usage, and economic power. While downloads are a reasonable proxy, several conclusions are phrased too strongly given the limitations of this signal. I recommend consistently softening claims or explicitly scoping them to “download-based estimates.”
* *Address limitations of country attribution*.
The current attribution of models to countries based on organizations is problematic for global companies. The authors should (i) discuss this limitation explicitly and (ii) clarify how it may affect the geopolitical conclusions. If possible, a more nuanced attribution (or sensitivity analysis) would strengthen the claims.
* *Justify or revise the download counting heuristic*.
The “one download per user per day” rule is not well motivated and introduces inconsistencies. The authors should either provide a stronger justification for why this approximates usage or include alternative counting schemes (e.g., stricter deduplication) and show that results are robust.
* *Reassess and better justify the rolling window filter*.
The 1-year rolling window is a central design choice, but it likely biases the results by undercounting sustained usage of older models. This is particularly concerning given the paper’s own observation that adoption lags behind development. At minimum, the authors should:
  * Provide a stronger justification for this assumption
  * Include sensitivity analyses (e.g., longer windows or no filtering)
  * Clearly discuss how this choice affects the interpretation of results
* *Discuss alternative usage signals*.
The paper should acknowledge and discuss complementary proxies for model adoption (e.g., mentions in academic publications, benchmark usage, citations). Even if not included in the analysis, this would better contextualize the limitations of the current approach.

**Suggested improvements:**
* *Improve figure readability*.
   * In Figure 2, plot the “Rest” category at the bottom to improve interpretability.
   * Add explicit subplot titles to Figures 2 and 3 to make them easier to read.
* *Minor writing and formatting issues*.
   * Fix typo: “has decline has” in Section 3.1
   * Add missing parentheses for citations in the last paragraph on page 4
   * Reduce overuse of “—” dashes throughout the paper

---

> ### Author Response · Authors · 2026-05-12
>
> We sincerely thank Reviewer ob6G for the thoughtful and encouraging review. We are grateful that they found the dataset comprehensive, the analysis timely and policy-relevant, the concentration metrics clear and interpretable, and the dataset/dashboard release valuable for future research. Their main concerns focus on proxy interpretation and methodological assumptions. We agree these should be clearer, and we have revised the paper to qualify claims, add sensitivity analyses, and improve figure readability.
>
> **Downloads as a proxy for usage and influence.**
> We agree that downloads should not be equated with deployment, model invocations, API usage, revenue, or total downstream adoption. We revised the paper to consistently use “download-based usage share” and “platform-mediated usage concentration,” and added explicit limitations in the Introduction, Methodology, Discussion, and Ethics Statement. Our revised framing is that downloads are the most comprehensive longitudinal signal available for open-model distribution and reuse, but remain an imperfect proxy.
>
> **Country attribution.**
> We agree that organization-headquarters attribution is imperfect, especially for global companies and distributed teams. We expanded the methodological caveats to state that country labels reflect organization headquarters, while unaffiliated users are intentionally not assigned countries for privacy. We also softened geopolitical claims so that China/US comparisons are explicitly scoped to our organization-headquarters, download-based attribution.
>
> **Download counting heuristic.**
> We clarified that the “one download per user per model per day” rule is the privacy-preserving deduplication used by Hugging Face’s application team. It reduces repeated automated pulls while preserving repeated activity across days, which is often the strongest observable signal of ongoing use available without retaining user-level identifiers. We also explicitly acknowledge that alternative signals and stricter deduplication schemes would be valuable complementary evidence.
>
> **Rolling Window Filter.**
> We added an appendix sensitivity analysis comparing the main one-year window with no recency cap and a stricter six-month cap. The core qualitative results hold: concentration declines through the diffusion period and rises again in 2025. The no-cap variant visibly overweights older BERT/CLIP/embedding-era models, consistent with automated dependency downloads, while the six-month cap makes recent shifts sharper. We retained one year as a conservative middle ground, but are open to suggestions! We do our best to justify these values based on discussions with the Hugging Face team and the observed effects of automated download pipelines, but acknowledge more readily in the paper there is no gold answer---and these metrics can be adjusted to measure different things.
>
> **Alternative usage signals.**
> We added discussion of complementary adoption proxies, including scholarly mentions, benchmark submissions, GitHub dependencies, API traces where available, and qualitative deployment studies. We now explicitly frame these as important future triangulation signals rather than substitutes for the longitudinal Hub-wide download record.
>
> **Figure readability.**
> We added explicit subplot titles to Figures 2 and 3 to make the panels easier to parse. We also tested moving the “Rest” category to the bottom of Figure 2, but after visual inspection found that it made the developer-share plot less readable, so we reverted to the original ordering.
>
> **Minor writing and formatting issues.**
> Thank you for catching these. We fixed the typos you caught, and reduced em-dash constructions. Note that the authors were using em-dashs long before they became a signal of automated tools. The paper was entirely human written (hence the typos you caught)!
>
> **Broader impact.**
> We agree that proxy-based measures can be overinterpreted in policy or geopolitical contexts. We strengthened the Ethics Statement to warn that download shares should not be interpreted as direct measures of geopolitical strength, commercial value, downstream societal impact, or real-world deployment.
>
> We sincerely thank the reviewer again for their constructive feedback. We believe these revisions make the paper more careful and stronger while preserving the central empirical contribution. We hope the changes address the reviewer’s concerns, and we would welcome any further suggestions they would like us to incorporate.

---

### Review · Reviewer_Cx5B · 2026-04-25

**Summary Of Contributions:**

This paper studies the Hugging Face Model Hub from June 2020 to August 2025. It analyzes 851K public models and 2.2B deduplicated downloads, and uses a Head Sample of 2,875 high-download models for detailed annotations. The paper studies model, developer, and national concentration using download shares, HHI, and Gini coefficients. It argues that open model usage shifted from US industry-dominated embedding models, to diffusion model communities, to a recent period with larger, multimodal, quantized models and rising Chinese developer share.

**Strengths:**

**S1**: The dataset itself is a substantial contribution: 5 years of weekly download logs for 851k models with rich metadata, enabling longitudinal study of the open model ecosystem at a scale not previously available.

**S2**: The three-era periodization (Foundation Embeddings, Generative Diffusion, Sino-Multimodal) provides a useful and empirically grounded framework for understanding structural shifts in the ecosystem.

**S3**: The paper combines multiple levels of analysis (model, developer, nation) with established economic concentration metrics, giving a more complete picture than prior static snapshots.

**Weaknesses:** see W1-W6.

**Additional Comments:**

**C1**: A related data-management perspective on ML assets could be relevant here: Wang et al., "ML-Asset Management: Curation, Discovery, and Utilization," *PVLDB* 18.12, 2025.

**C2**: (1) Section 1: "closets available proxy" should be "closest available proxy." (2) Figure 3 caption: "Hherfindahl-Hirschman" should be "Herfindahl-Hirschman." (3) Section 3: "over 60 of national market share" is missing a percent sign. Careful proofreading is needed throughout.

**Audience:**

Yes

**Audience Explanation:**

The open model ecosystem is of broad interest to the ML community. Tracking how model usage, developer composition, and technical characteristics evolve over time on Hugging Face is relevant to researchers working on open-source ML, AI governance, and the science of science. The released dataset and dashboard could serve as useful infrastructure for future empirical work.

**Broader Impact Concerns:**

**I1**: The paper frames AI development through a geopolitical lens (US vs. China), which risks reinforcing nationalistic narratives about AI competition. The authors acknowledge this in the Ethics Statement but the main text still leans heavily into this framing.

**I2**: The "open-source compliance" framing is sensitive. The paper mainly measures training data disclosure and availability, not full legal or governance compliance with open-source AI definitions.

**Claims And Evidence:**

No

**Claims Explanation:**

**W1**: Sections 2, 3, and 6 treat Hugging Face downloads as evidence of market share, economic power, and influence. The data only measure filtered download events on one platform. The paper does not validate this proxy against deployment, API usage, model invocations, revenue, or downstream adoption.

**W2**: The longitudinal framing has a survivorship and metadata timing problem. Appendix A.1 says the download data only cover models that were publicly available at the time of data collection, so models that were removed, made private, or otherwise unavailable are excluded from the historical record. Appendix A.2 also appears to merge historical downloads with current model metadata and current model cards, which is especially problematic for claims about changing licenses, gating, data disclosure, and openness over time.

**W3**: The main concentration narrative depends on two under-specified design choices. The one-year Rolling Window Filter is central to the leaderboards and temporal share analysis in Figures 1 and 2, but the paper does not show whether the trends hold under other cutoffs. Recursive Model Attribution is also not formalized enough for multi-parent models, long derivation chains, or missing base-model links.

**W4**: Most attribute claims in Section 4 and Table 1 come from the Head Sample, 2,875 top-200-per-week models covering 49.6% of downloads, while the broader Population Sample covers 851k models and 97.6% of downloads. This sample is explicitly biased toward popular models, yet the paper describes ecosystem-wide shifts in architecture, modality, transparency, gating, and training methods. The annotation quality claim of "90%+ accuracy" has no reported validation set, inter-annotator agreement, adjudication protocol, or error bars, so the Head Sample results are hard to audit.

**W5**:  The national power analysis is too coarse for the geopolitical claims. Appendix A.3 assigns countries from organization headquarters, labels individual contributors as "Individual", and puts some groups into "Online" or "International". This loses information about multinational labs, distributed teams, model uploaders who repackage another group's work, and users whose countries are intentionally not annotated. Claims such as "China surpassed the United States" need stronger caveats because a large share of recent downloads is assigned to Unaffiliated User or International/Online.

**W6**: The statistical treatment in Table 1 is weak. The table is download-weighted, but the paper does not clearly specify the unit used in the chi-squared tests. Weekly downloads from the same model are not independent observations. With huge counts, p < 0.001 is not informative, and the table reports no confidence intervals, multiple-comparison correction, or uncertainty from annotation and model-size estimation.

**Requested Changes:**

Address or explain W1-W6.

---

> ### Author Response · Authors · 2026-05-12
>
> We sincerely thank Reviewer Cx5B for the thoughtful and detailed review. We are especially grateful that they recognized the dataset as a substantial contribution, found the three-era periodization useful and empirically grounded, and appreciated the multi-level analysis of models, developers, and nations using established concentration metrics. Their concerns largely focus on scope, proxy validity, attribution assumptions, and statistical interpretation. We agree these points are important, and we have revised the paper to make the claims more precise, add sensitivity analyses, and expand the limitations.
>
> **Downloads as a proxy for usage, economic power, and influence.**
> We agree and have softened the framing throughout the paper. We now consistently describe our measurements as “download-based usage share” or “platform-mediated usage concentration,” rather than direct market share or economic power. We also added text in the Introduction, Methodology, Discussion, and Ethics Statement clarifying that downloads do not directly measure deployments, API calls, revenue, model invocations, or downstream adoption. Our revised claim is that downloads are the most comprehensive longitudinal signal currently available for open-model distribution and reuse, but remain an imperfect proxy.
>
> **Survivorship and current-metadata timing.**
> We agree this limitation should be more visible. We added explicit caveats in Appendix A.1 and the Ethics Statement that our historical record only includes models publicly available at collection time, so removed/private models are excluded. We also clarified that metadata such as licenses, gating, model cards, and training-data disclosure may reflect current public metadata rather than the exact historical state at the time of each download.
>
> **Rolling Window Filter and Recursive Model Attribution.**
> We added an appendix sensitivity analysis comparing the main one-year window with no recency cap and a stricter six-month cap. The qualitative trend remains: early concentration declines through the diffusion period and rises again in 2025. The no-cap variant visibly overweights older BERT/CLIP/embedding-era models, consistent with stale automated dependency downloads. In discussion with the HuggingFace team, we are told most such automated download pipelines do not correspond to actual downstream usage. For privacy reasons, however, they cannot share or reveal *who* these downloads are from.
>
> We also formalized Recursive Model Attribution: for a downloaded derivative model, we recursively follow parent links to the root model and attribute downloads to the root author/country. In the rare model merging cases when multiple parents are listed, we use the first documented parent.
>
> **Head Sample scope and annotation quality.**
> We agree the original text sometimes read too broadly. We revised Section 4 to state that Head Sample results are download-weighted trends among high-usage models, not unweighted prevalence across the full Hub. We also clarified the annotation quality statement: annotators completed multiple rounds of training guided by expert annotators before real annotation, and an expert NLP researcher reviewed a random sample of about 10% of all annotator labels. All fields were at least 90% accurate in this expert review, including model architecture, which was the lowest-scoring category. We added these details to the appendix and are confident in the annotation quality.
>
> **National attribution and geopolitical interpretation.**
> We agree and have strengthened the caveats. We now state that country labels are organization-headquarters attributions, not a complete geopolitical accounting of development labor. We explicitly note that multinational companies, distributed teams, repackaged uploads, unaffiliated users, and International/Online groups limit national interpretation. We also softened “China surpassed the United States” to “in our organization-headquarters, download-based attribution.”
>
> **Statistical treatment in Table 1.**
> We revised the Table 1 caption and surrounding framing to emphasize descriptive effect sizes rather than large-count significance. We now state that the table is a download-weighted descriptive summary over the Head Sample and that statistical screens use model-level contingency tests over unique annotated models, not weekly download rows. The interpretation is now based on magnitude and direction of changes rather than the absolute p-values.
>
> **Broader impacts feedback.**
> We agree that geopolitical framing and “open-source compliance” language require care. We carefully adjusted the geopolitical language throughout, and added a stronger Ethics Statement warning against interpreting download shares as direct measures of geopolitical strength or commercial value. We also replaced “open-source compliance” framing with the narrower and more accurate “training-data transparency” and “open-source AI documentation criteria.”

---

> > ### Author Response · Authors · 2026-05-12
> >
> > **Additional comments and typos.**
> > Thank you for the careful proofreading and related-work suggestion. We fixed all typos and inlcuded your minor suggestions. We also added Wang et al., “ML-Asset Management: Curation, Discovery, and Utilization,” PVLDB 2025, to Related Work.
> >
> > We sincerely thank the reviewer again for the constructive and detailed feedback. We believe these amendments make the paper more precise, better scoped, and more useful to readers. We hope these changes thoroughly address the reviewer’s concerns, and we would welcome any additional changes they would like to see.

---

> > > ### Comment · Reviewer_Cx5B · 2026-06-09
> > >
> > > Thank you for the detailed rebuttal. I appreciate that the authors clarified the scope of the download-based claims, added caveats and sensitivity analyses, and softened the framing. Some limitations remain around proxy validity and attribution uncertainty, but the revision is better scoped.
> > >
> > > I also want to emphasize that I find the contribution meaningful for the ML community: beyond the raw model counts, the paper provides a large-scale longitudinal dataset and useful measurement infrastructure for studying open model ecosystems, model reuse, and AI governance.

---

### Review · Reviewer_1mKp · 2026-05-10

**Summary Of Contributions:**

This paper is trying to do a survey of the kinds of models on HuggingFace.
It tracks various parameters like whether the models are open or closed, their size and country of origin.

**Audience:**

No

**Audience Explanation:**

A sociological survey of models is of negligible interest to the audience of a top journal in ML.

This is not ML research.

**Claims And Evidence:**

Yes

**Claims Explanation:**

The paper merely does a counting of the models on HuggingFace.

**Requested Changes:**

There is no viable way to improve this paper.
In my view this topic is out of scope for TMLR.

---

> ### Author Response · Authors · 2026-05-12
>
> We thank Reviewer 1mKp for raising the scope concern, and we are eager to engage with this and other constructive feedback. However, we are a little disappointed by the framing of our work as "merely doing a counting of the models on Hugging Face". We find this to be dismissive of a significant and genuine effort we've made towards open scientific questions, even if you question the fit to TMLR.
>
> Our contribution is a longitudinal measurement framework, dataset, and infrastructure for understanding how open ML systems are produced, reused, concentrated, and transformed over time. We believe there is clear precedent that this is a good fit for TMLR, as we outline below, and we hope to convince you of this.
>
> **Why we believe this is in scope for TMLR.**
> TMLR explicitly includes work on "new approaches for analysis, visualization, and understanding of artificial or biological learning systems" (including ecosystems of AI models), and "surveys that draw new connections, highlight trends, and suggest new problems in an area." TMLR has accepted (and even bestowed "Featured Certifications" on) closely related contribution types: human-collected scientific measurement datasets [1], transparency indices that score AI ecosystem actors against explicit criteria [2,3], open-weight AI ecosystem/risk surveys [4], resource collections in the AI ecosystem [5], and surveys of AI systems/evaluation tools [6]. None of these would be considered "ML research" by your standards. Our paper offers a combination of these contribution types, all well validated and widely accepted at TMLR:
>
> - **A dataset and infrastructure for studying the open model ecosystem.** We release a longitudinal dataset covering 851K public models, 2.2B deduplicated downloads, model lineage/derivation graphs, architectures, modalities, access restrictions, training-data transparency, and other rich metadata, together with reproducible code and a dashboard for future scientific study.
> - **A survey of open-model ecosystem measurement.** We synthesize prior work and relevant methods for measuring model adoption, reuse, transparency, concentration, and ecosystem health, situating Hugging Face as a measurable infrastructure layer for studying modern ML systems.
> - **Methods and metrics for measuring AI usage concentration.** We operationalize download-based usage share, HHI/Gini concentration, Recursive Model Attribution, and rolling-window sensitivity analyses to estimate how influence and reuse concentrate across models, developers, and country attributions.
> - **An ecosystem audit with empirical findings.** We apply these methods to document major shifts in open-model reuse: embedding-era dominance, diffusion-community growth, reconsolidation around large multimodal and quantized models, rising MoE usage, changing developer composition, and declining training-data transparency.
>
> These contributions fit cleanly among those that TMLR has accepted and awarded in the past.
>
>
> **Audience interest.**
> TMLR asks whether "at least some individuals in TMLR's audience" would be interested. We see clear evidence that researchers who build, release, audit, benchmark, or depend on open models have a direct interest in understanding which models dominate usage, how reuse concentrates, and how model properties evolve.
>
> The review does not identify a technical weakness or actionable revision. We are nevertheless excited to engage on questions of scope or how we can revise the paper to improve it in the eyes of the reviewer.
>
> **References.**
>
> [1] *A Gold Standard Dataset for the Reviewer Assignment Problem*, TMLR 2025. https://openreview.net/forum?id=XofMHO5yVY
>
> [2] *The 2023 Foundation Model Transparency Index*, TMLR 2025. https://openreview.net/forum?id=x6fXnsM9Ez
>
> [3] *The 2024 Foundation Model Transparency Index*, TMLR 2025. https://openreview.net/forum?id=38cwP8xVxD
>
> [4] *Open Technical Problems in Open-Weight AI Model Risk Management*, TMLR 2026. https://openreview.net/forum?id=8QyGLnFkzc
>
> [5] *The Responsible Foundation Model Development Cheatsheet: A Review of Tools & Resources*, arXiv 2024. https://arxiv.org/abs/2406.16746
>
> [6] *Measuring Data Science Automation: A Survey of Evaluation Tools for AI Assistants and Agents*, TMLR 2025. https://openreview.net/forum?id=MB0TCLfLn1

---

> ### Comment · Reviewer_1mKp · 2026-06-24
> **Thanks!**
>
> Thanks for collating these references to these previous TMLR papers which you claim to be of the same genre.
>
> I think these references you brought forth are like position papers in ML.
> They are surveying the state of a technical issue in ML.
>
> That is not what this paper is doing - here the question being looked into is purely a sociological one and not a scientific one.